# GRIFFIN: Effective Token Alignment for Faster Speculative Decoding

**Shijing Hu**[1]  **Jingyang Li**[2]  **Xingyu Xie**[2]  **Zhihui Lu**[1*]  **Kim-Chuan Toh**[2]  **Pan Zhou**[3]

[1]Fudan University    [2]National University of Singapore    [3] Singapore Management University

sjhu24@m.fudan.edu.cn    li_jingyang@u.nus.edu    xyxie@pku.edu.cn

lzh@fudan.edu.cn    mattohkc@nus.edu.sg    panzhou@smu.edu.sg

## Abstract

Speculative decoding accelerates inference in large language models (LLMs) by generating multiple draft tokens simultaneously. However, existing methods often struggle with token misalignment between the training and decoding phases, limiting their performance. To address this, we propose GRIFFIN, a novel framework that incorporates a token-alignable training strategy and a token-alignable draft model to mitigate misalignment. The training strategy employs a loss masking mechanism to exclude highly misaligned tokens during training, preventing them from negatively impacting the draft model's optimization. The token-alignable draft model introduces input tokens to correct inconsistencies in generated features. Experiments on LLaMA, Vicuna, Qwen and Mixtral models demonstrate that GRIFFIN achieves an average acceptance length improvement of over 8% and a speedup ratio exceeding 7%, outperforming current speculative decoding state-of-the-art methods. Our code and GRIFFIN's draft models will be released publicly in https://github.com/hsj576/GRIFFIN.

## 1 Introduction

Large Language Models (LLMs) like GPT-4 [1] and LLaMA [2, 3] have shown impressive capabilities in diverse domains, including dialogue [4] and code generation [5]. However, the standard autoregressive decoding of LLMs generates tokens sequentially, with each token requiring a full forward pass through the entire model. Given the large size of LLMs, this process is both computationally expensive and time-consuming, posing challenges for latency-sensitive applications. To accelerate generation, speculative decoding [6, 7] has become widely adopted and shown significant speed improvements. It leverages a lightweight draft model to propose multiple tokens, verifies them in parallel with the target LLM, and accepts those aligned with the target's predictions. This enables multi-token generation per forward pass of the target LLM, substantially reducing latency.

However, the efficiency of speculative decoding depends critically on achieving a high acceptance rate for draft tokens, while also minimizing the computational cost of generating them. Recent methods like EAGLE [8, 9] and Medusa [10] address this by utilizing shallow-layer hidden states of the target LLM to guide draft model's token predictions. Despite their improved efficiency, these methods face a fundamental limitation: misalignment between the training and decoding processes. During training, the draft model uses features from the target model and ground-truth tokens from training data, whereas in decoding, it relies on its own generated features and previously generated draft tokens. This discrepancy introduces two key issues: (1) feature misalignment, where the features generated by the draft model during decoding diverge from those features used during training, and (2) token misalignment, where ground-truth tokens are replaced by draft tokens, often compounding

---

[*]Corresponding author.

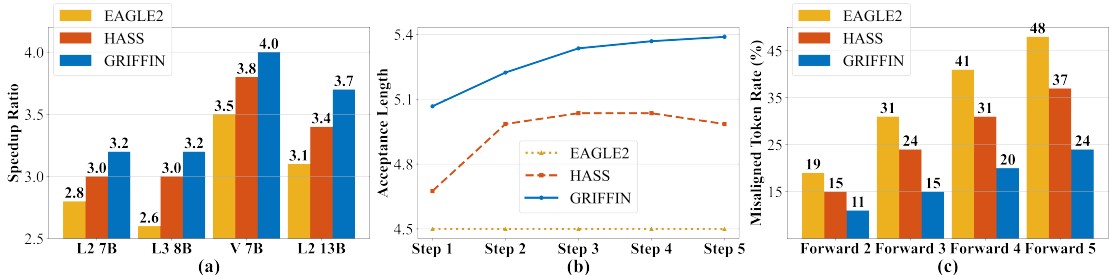

Figure 1: Comparison between our GRIFFIN, EAGLE2, and HASS. (a) Speedup ratio comparison. (b) Acceptance length under different training steps, in which "Step $n$" denotes aligning draft model for $n$ times in training. (c) Misaligned token rate under different forward passes in each drafting-verification cycle, where "Forward $n$" denotes forwarding $n$ passes to generate $n$ draft tokens.

errors over multiple steps. These misalignments, akin to exposure bias [11, 12], significantly degrade the acceptance rate of draft tokens and thus impair the overall speedup performance.

Efforts to address feature misalignment like HASS [13] use draft model's features to replace target model's features during training. While aligning training with decoding, it neglects token misalignment which is particularly problematic in decoding. Errors from earlier decoding steps propagate and amplify, further exacerbating token misalignment. For instance, as shown in Fig. 1 (c), EAGLE2 suffers from a token misalignment rate of 48% during training, resulting in suboptimal acceptance lengths and limiting its effectiveness. Similarly, while mitigating feature misalignment, HASS sees token misalignment escalate to 37% in later training steps, rendering it ineffective for deeper multi-forward harmonized training, e.g., training step $n \geq 3$, as shown in Fig. 1 (b).

**Contributions.** We propose GRIFFIN, a novel speculative decoding framework that—unlike prior work—explicitly identifies and addresses the previously unobserved token misalignment issue, alongside feature misalignment. It introduces two core innovations: a token-alignable training strategy and a token-alignable draft model, working together to significantly boost decoding efficiency.

Firstly, to mitigate token misalignment during training, GRIFFIN employs a dynamic loss masking mechanism that selectively backpropagates only through aligned tokens—defined as those whose ground-truth tokens appear in the draft model's top-$k$ predictions. This not only minimizes the disruptive effect of highly misaligned tokens, but also harmonizes training and decoding since draft trees in decoding builds upon top-$k$ predictions rather than exact matching to the highest-probability token. Unlike prior works [8, 9, 13] which train on exact targets, GRIFFIN embraces approximation while preserving signal fidelity, improving both alignment and generalization across decoding steps.

Secondly, to further reduce token misalignment, GRIFFIN designs a *token-alignable draft model* by incorporating the architectural innovation of Token-Guided Fusion (TGF) into draft model in EAGLE [8]. TGF performs a two-step fusion to refine feature representations and mitigate inconsistencies between the draft and target models. By incorporating input tokens twice—initially with features and later to refine them—our TGF module ensures that the draft model produces features more closely aligned with the target model, reducing feature and token misalignment.

These two components are mutually reinforcing. The token-alignable draft model reduces misalignment, increasing the number of aligned tokens available for effective training. In turn, the training strategy ensures that these aligned tokens contribute meaningfully to model optimization. Crucially, our approach is the first to expose and directly address token misalignment—an uncharted limitation in speculative decoding that hampers draft token acceptance and decoding speed. As shown in Fig. 1 (c), GRIFFIN consistently maintains a much lower token misalignment rate compared to EAGLE2 and HASS across multiple forward steps. This yields longer accepted token sequences and greater speedups, particularly in deeper harmonized training settings where previous methods degrade.

Experimental results show GRIFFIN's superior performance over state-of-the-arts (SoTAs) across diverse tasks, including dialogue (MT-Bench [4]), code generation (HumanEval [5]), and mathematical reasoning (GSM8K [14]). For example, Fig. 1 (a) and (b) show that on LLaMA2-7/13B, LLaMA3-8B, and Vicuna-7B, GRIFFIN improves the average acceptance length by 20% over EAGLE2 and 8% over HASS, while delivering a speedup ratio of 18% over EAGLE2 and 7% over HASS.

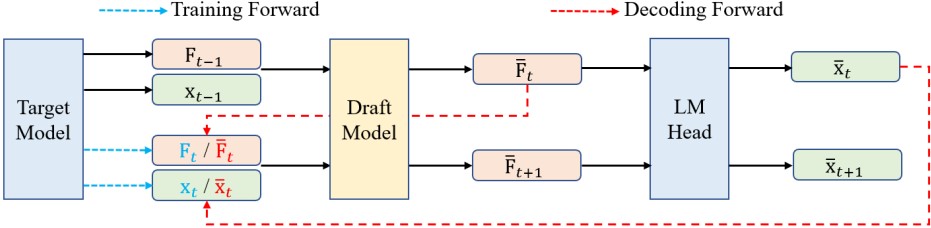

Figure 2: Token and feature misalignment in EAGLE.

## 2   Related Work

Speculative decoding [15–19] accelerates LLM inference by dividing each decoding step into a draft stage and a verification stage. Existing methods differ primarily in their draft model architectures or strategies, each addressing specific challenges in speculative decoding.

Several methods enhance draft quality through context retrieval, e.g., PLD [20], Lookahead [21], and CLLMs [22], which rely on prompt-based retrieval from similar contexts. However, their effectiveness is limited when relevant context is scarce or unavailable. Tree-based verification approaches like Sequoia [23] and SpecExec [24] use hierarchical structures to improve verification but incur high computational overhead, making them unsuitable for latency-sensitive scenarios. Other works, including REST [25] and Ouroboros [26], reuse previous outputs or databases to guide drafting but are constrained by the quality and accessibility of external resources. Chimera [27] and Glide [28] enhance token quality by integrating target model into draft model with extra computational cost.

Lightweight draft models have also been explored to improve efficiency. Medusa [10] employs MLPs for parallel candidate prediction, while Hydra [29] and Recurrent Drafter [30] use RNN-based models for regressive generation. EAGLE [8] and EAGLE2 [9] introduces a transformer decoder for autoregression over feature sequences, balancing accuracy and complexity. FSPAD [31] constructs input sequences tailored for lightweight draft model predictions and introduces specialized training methods to improve draft quality. Additionally, methods like HASS [13] address feature misalignment during training and decoding but do not fully resolve token-level misalignment. In contrast, this work focuses on addressing token misalignment, a critical challenge in speculative decoding. We propose the GRIFFIN framework, which introduces a token-alignable training strategy and a token-alignable draft model. By tackling this issue, GRIFFIN improves both acceptance length and speedup ratio, offering a complementary perspective to existing methods.

## 3   Motivation: Token Misalignment

Speculative decoding [6, 7] accelerates text generation by employing a "draft-and-verify" strategy. Per cycle, a lightweight draft model $\mathcal{M}$ first generates multiple tokens through multiple forward passes, and a stronger target model $\mathcal{T}$ then verifies and accepts a subset in a single forward pass.

EAGLE [8] extends this paradigm by shifting autoregression from the token level to the feature level. As shown in Fig. 2, instead of predicting tokens directly, the draft model generates intermediate hidden state features that approximate those from the final layer of the target model $\mathcal{T}$—just before the language modeling (LM) head $\mathcal{H}$. At time step $t$, let $\mathbf{x}_t$ and $\bar{\mathbf{x}}_t$ denote the $t$-th ground-truth and draft tokens, and $\mathbf{F}_t$ and $\bar{\mathbf{F}}_t$ their respective hidden features from $\mathcal{T}$ and $\mathcal{M}$. During training, as illustrated in Fig. 2, the draft model uses $\mathbf{x}_t$ and $\mathbf{F}_t$ to predict $\bar{\mathbf{x}}_{t+1}$ and $\bar{\mathbf{F}}_{t+1}$. However, during decoding, the draft model must rely solely on previously generated tokens $\bar{\mathbf{x}}_t$ and features $\bar{\mathbf{F}}_t$—without access to $\mathbf{x}_t$ or $\mathbf{F}_t$—since the target model is invoked only once per cycle after all draft tokens are produced. This discrepancy introduces two fundamental issues: 1) **feature misalignment** where during decoding, for prediction, the draft model uses $\bar{\mathbf{F}}_t$ instead of $\mathbf{F}_t$ as in training; and 2) **token misalignment** where the tokens $\bar{\mathbf{x}}_t$ used in decoding differ from ground-truth tokens $\mathbf{x}_t$ seen during training.

Among these, token misalignment is particularly severe yet underexplored. Fig. 1(c) shows that for EAGLE2 and HASS, the rate at which $\bar{\mathbf{x}}_t \neq \mathbf{x}_t$—the token misalignment rate—increases sharply with the number of forward passes. For instance, EAGLE2 reaches a misalignment rate of 48% when

generating five draft tokens per cycle. Even HASS, which partially mitigates feature misalignment, still suffers from a 37% token misalignment rate. This degradation stems from error accumulation across passes, where early mistakes in token generation propagate and compound in subsequent steps.

Critically, high token misalignment undermines training effectiveness. As shown in Fig. 1(b), when the number of forward passes exceeds three in HASS, acceptance length plateaus—even with continued training. This suggests that draft models only generate fewer acceptable tokens, directly limiting decoding efficiency. Hence, solving token misalignment is not only important but necessary to unlock deeper multi-pass speculative decoding and greater speedups.

A seemingly simple fix—replacing $\mathbf{x}_t$ (ground-truth training tokens) with $\bar{\mathbf{x}}_t$ from the draft model during training—fails in practice. This is because 1) frameworks like EAGLE and HASS precompute and store $\mathbf{F}_t$ for all $\mathbf{x}_t$ before training which avoids the computational burden of regenerating training data; 2) swapping in $\bar{\mathbf{x}}_t$ leads to inconsistent input-feature pairs, which breaks the alignment needed for loss computation and degrades performance, as confirmed by Appendix. B in HASS. In fact, naive substitution significantly reduces acceptance length. In light of these challenges, we propose an effective solution to the token misalignment problem that preserves compatibility with existing training workflows and enables better alignment between training and decoding.

## 4 GRIFFIN: A Token-Alignable Framework

To address token misalignment challenges in Sec. 3, we propose GRIFFIN, a novel framework to mitigate token misalignment through two key components: 1) token-alignable training introduced in Sec. 4.1 and 2) a token-alignable draft model elaborated in Sec. 4.2.

### 4.1 Token-Alignable Training

At the core of GRIFFIN is a progressive training strategy that mirrors how the draft model operates during decoding. Instead of relying on ground-truth tokens and features at every step—an assumption that breaks down during inference—we gradually shift the model toward using its own outputs during training. This alignment is critical to mitigating token misalignment.

Concretely, GRIFFIN organizes training into multiple steps, where each training step $n$ involves draft model performing $n$ forward passes to predict $n$ future tokens and their corresponding features. With each additional pass, the model increasingly conditions on its own generated tokens and features from prior steps rather than ground-truth tokens and features from target model. This effectively aligns training process with decoding phase, as in training phase, the draft model simulates the similar input conditions encountered during decoding. Then we detail the first training step and its subsequent step.

**First Forward Pass** ($n = 1$): Like vanilla autoregressive generation, draft model $\mathcal{M}$ predicts draft tokens which are then fed into target model $\mathcal{T}$ to verify and accept. Specifically, at time step $t$, draft model $\mathcal{M}$ and LM head $\mathcal{H}$ in target model predicts the $t$-th feature embedding $\bar{\mathbf{F}}_t$ and draft token $\bar{\mathbf{x}}_t$:

$$\bar{\mathbf{F}}_t = \mathcal{M}(\mathbf{x}_{1:t-1}, \mathbf{F}_{1:t-1}), \qquad \bar{\mathbf{x}}_t = \mathcal{H}(\bar{\mathbf{F}}_t), \tag{1}$$

where $\mathbf{x}_{1:t-1}$ denotes the token sequence $\{\mathbf{x}_i\}_{i=1}^{t-1}$ from training dataset and $\mathbf{F}_{1:t-1}$ are the feature embedding sequence $\{\mathbf{F}_i\}_{i=1}^{t-1}$ generated by target model $\mathcal{T}$.

Token misalignment arises only from the second forward pass onward. In the first pass, both training and decoding share the identical ground-truth prefix $\mathbf{x}_{1:t-1}$, so all predicted tokens $\bar{\mathbf{x}}_t$ are perfectly aligned. Therefore, no masking is needed in the first pass. The first-pass loss at step $t$ is

$$\mathcal{L}_{\mathcal{M}}^{(1)} = \sum_{t=1}^{l} \ell(\bar{\mathbf{x}}_t, \mathbf{x}_t, \bar{\mathbf{F}}_t, \mathbf{F}_t), \tag{2}$$

where $\ell$ combines a feature-level loss, i.e., the $\ell_1$ distance between $\bar{\mathbf{F}}_t$ and $\mathbf{F}_t$, and a token-level loss, namely, cross-entropy between $\bar{\mathbf{x}}_t$ and $\mathbf{x}_t$. The detail implementation of $\ell$ is summarized in Appendix. C.

$n$**-th Forward Pass** ($n \geq 2$)**.** Draft model $\mathcal{M}$ would predict $n$ draft tokens at the $n$-th forward pass. From the second forward pass, during decoding, speculative decoding may reject a draft token $\bar{\mathbf{x}}_t$, in which case *all* later tokens $\bar{\mathbf{x}}_{t+1}, \bar{\mathbf{x}}_{t+2}, \ldots$ in that draft sequence are also discarded. So, during

training, if $\bar{\mathbf{x}}_t$ is unpredictable (rejected), the subsequent draft tokens $\bar{\mathbf{x}}_{t+1}, \bar{\mathbf{x}}_{t+2}, \ldots$ in this draft sequence are **misaligned tokens**. Training on those misaligned token provides no useful signal.

To address token misalignment challenge, we introduce a novel token-alignable training strategy that aligns draft model's training with its multi-pass behavior during decoding. Unlike prior approaches like EAGLE which use only top-1 predictions during training, our method incorporates the tree-structured decoding process directly into learning by supervising on top-$k$ predictions. In EAGLE's decoding, each forward pass of draft model generates a top-$k$ list of candidate tokens at each time step, forming a tree where alternative branches can be explored if the top-1 token is rejected. To match this, GRIFFIN considers a draft token $\bar{\mathbf{x}}_t$ to be predictable if the ground-truth token $\mathbf{x}_t$ appears within its top-$k$ predictions Top-$k(\bar{\mathbf{x}}_t)$. This ensures that training reflects the decoding phase, where any top-$k$ token may be valid. Accordingly, we introduce a binary predictable mask $\bar{\mathbf{m}}_t \in \{0, 1\}$, where $\bar{\mathbf{m}}_t = 1$ if $\mathbf{x}_t \in$ Top-$k(\bar{\mathbf{x}}_t)$, and $\bar{\mathbf{m}}_t = 0$ otherwise. Since the current draft token $\bar{\mathbf{x}}_t$ is decided by previous predicted draft tokens $\bar{\mathbf{x}}_{t-n+1:t-1}$ predicted in earlier $(n-1)$ forward passes, then if any draft token in $\bar{\mathbf{x}}_{t-n+1:t-1}$ is unpredictable, the draft token $\bar{\mathbf{x}}_t$ would likely be misalignment. To prevent the model from being penalised for such inevitably rejected positions, we introduce a cumulative binary alignment mask $\mathbf{m}_t$ adjusted by predictable masks $\bar{\mathbf{m}}_{t-n+1:t-1}$ of draft tokens $\mathbf{x}_{t-n+1:t-1}$:

$$\mathbf{m}_t = \prod_{i=t-n+1}^{t-1} \bar{\mathbf{m}}_i. \tag{3}$$

These masks indicate whether a token should contribute to the training loss, ensuring consistency between training and inference. Next, to further ensure alignment between training and decoding, we replace target-model features $\mathbf{F}_{t-n+1:t-1}$ with draft-model-generated features $\bar{\mathbf{F}}_{t-n+1:t-1}$ from earlier passes. Then the draft model $\mathcal{M}$ and the LM head $\mathcal{H}$ are used to generate the feature $\bar{\mathbf{F}}_t$ and the draft token $\bar{\mathbf{x}}_t$:

$$\bar{\mathbf{F}}_t = \mathcal{M}(\mathbf{x}_{1:t-1}, \mathbf{F}_{1:t-n}, \bar{\mathbf{F}}_{t-n+1:t-1}), \quad \bar{\mathbf{x}}_t = \mathcal{H}(\bar{\mathbf{F}}_t). \tag{4}$$

Then, we define the following training loss to train the draft model $\mathcal{M}$:

$$\mathcal{L}_{\mathcal{M}}^{(n)} = \frac{1}{\sum_{t=1}^{l} \mathbf{m}_t} \sum_{t=1}^{l} \mathbf{m}_t \ell(\bar{\mathbf{x}}_t, \mathbf{x}_t, \bar{\mathbf{F}}_t, \mathbf{F}_t), \tag{5}$$

GRIFFIN's training strategy differs from priors like EAGLE and HASS that rely on ground-truth tokens during training. By progressively adapting draft model to operate under its own predictions and aligning its training with decoding, GRIFFIN addresses token misalignment issue via introducing top-$k$ alignment masks, self-conditioning through generated tokens, and mask propagation.

## 4.2 Token-Alignable Draft Model

To enhance draft token accuracy and effectively address token misalignment, we propose a token-alignable draft model which systematically resolves feature inconsistency issues overlooked by prior draft models. While our architecture builds on EAGLE's draft model, it introduces two key extra modules: Token-Guided Fusion (TGF) and Token-Enhanced Head (TEH). As shown in Fig. 3(a), we insert TGF module before the autoregressive layer to fuse input features $\mathbf{F}_t$ and tokens $\mathbf{x}_t$. After autoregression, we use TEH module, a dual-head design inspired by prior work [31], to output 1) a predict feature $\bar{\mathbf{F}}_{t+1}^{P}$ for token prediction and 2) a regress feature $\bar{\mathbf{F}}_{t+1}^{R}$ to feed subsequent forward passes. Accordingly, TEH can separate and decouple the conflicting objectives of token prediction and feature generation within the draft model, improving draft token accuracy. Our ablation in Appendix. B confirm its effectiveness.

TGF module is designed to address a core challenge: feature representations in draft models often fail to match those of the target model, even after extensive training. Since feature-level losses can't be minimized to zero in practice, this gap leads to persistent misalignments, leading to the misaligned features between draft and target models which impairs draft token's accuracy. TGF tackles this by prioritizing token embeddings in the fusion process, guiding the feature generation toward better consistency with the target model. As illustrated in Fig. 3(b), TGF operates in three steps:

**(1) Embedding Fusion** in Fig. 3 (b-i). Given input feature $\mathbf{F}$ and token embedding $\mathbf{x}$ (both in $\mathbb{R}^d$), we concatenate them and use a lightweight MLP to project the result back to $\mathbb{R}^d$:

$$\mathbf{h} = \mathcal{C}(\mathbf{F}, \mathbf{x})\mathbf{W}_{\mathtt{m}} + \mathbf{b}_{\mathtt{m}}. \tag{6}$$

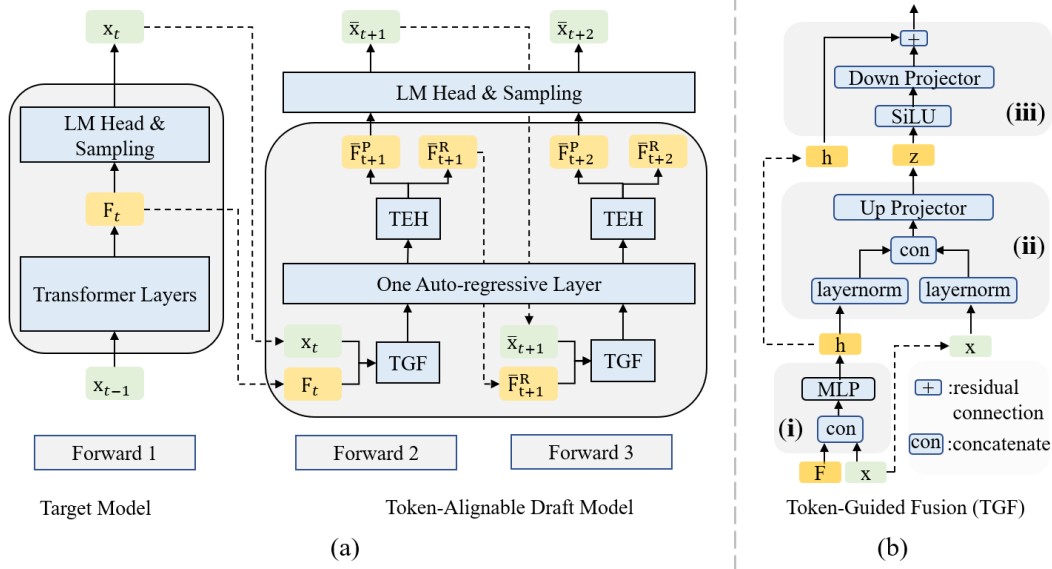

Figure 3: Structure of GRIFFIN's darft model. (a) Token-Alignable Draft Model. (b) TGF module. The diagram depicts the shared architecture used in both training and decoding phases—arrows indicating token flow correspond to valid data dependencies in both regimes.

Here, $\mathbf{W_m} \in \mathbb{R}^{2d \times d}$ and $\mathbf{b_m} \in \mathbb{R}^d$ are the MLP weights and bias, and $\mathcal{C}(\cdot, \cdot)$ is the concatenation operator. This produces a unified feature that blends both token and feature information.

**(2) Feature Normalization and Expansion** in Fig. 3 (b-ii). We apply layer normalization to both $\mathbf{h}$ and $\mathbf{x}$, then concatenate and expand the dimension to $4d$ using an Up Projector (a linear layer):

$$\mathbf{z} = \mathcal{C}(\mathcal{N}(\mathbf{h}), \mathcal{N}(\mathbf{x}))\mathbf{W_u} + \mathbf{b_u}, \tag{7}$$

with $\mathbf{W_u} \in \mathbb{R}^{2d \times 4d}$ and $\mathbf{b_u} \in \mathbb{R}^{4d}$. Here, $\mathcal{N}(\cdot)$ denotes layer normalization. Operating in a higher-dimensional space enables the model to disentangle and align more complex token-feature relationships. This $4d$ expansion aligns with the intermediate size used in many transformer FFNs, and our ablations in Sec. 5.2, Appendix A.2 and A.3 confirm its effectiveness.

**(3) Refinement and Stabilization** in Fig. 3 (b-iii). We apply a SiLU nonlinearity $\sigma$ to $\mathbf{z}$ and project it back to $\mathbb{R}^d$ using a Down Projector (a liner layer). A residual connection with $\mathbf{h}$ stabilizes training:

$$\mathbf{o} = \sigma(\mathbf{z})\mathbf{W_d} + \mathbf{b_d} + \mathbf{h}, \tag{8}$$

where $\mathbf{W_d} \in \mathbb{R}^{4d \times d}$ and $\mathbf{b_d} \in \mathbb{R}^d$. The nonlinearity enriches expressiveness, and the residual addition preserves essential fused information.

By explicitly integrating token embeddings into feature fusion, TGF ensures that generated features better reflect token distribution of target model. Combined with TEH, it enables draft model to generate more accurate draft tokens and features, crucial for mitigating misalignment in multi-pass decoding. This token- and feature-aware design is a key innovation that differs GRIFFIN from priors.

## 5   Experiments

Representative LLMs, including LLaMA2-Chat 7B/13B, LLaMA3-Instruct 8B/70B [3], Vicuna-1.5 7B [32], Qwen2-Instruct 7B [33] and Mixtral-8x7B-Instruct-v0.1 [34], are tested. For consistency, all inference runs use one NVIDIA A100 80G GPU, except for LLaMA3-70B and Mixtral-8x7B, which require two GPUs. Vanilla auto-regressive decoding is used as the baseline, serving as the benchmark for speedup ratios (1.00x). We compare GRIFFIN against recent SoTA speculative decoding methods, including SPS (standard speculative sampling with its draft model being Vicuna-68M) [6], PLD [20], Lookahead [21], Medusa [10], EAGLE [8], EAGLE-2 [9], FSPAD [31], and HASS [13]. We follow priors and train our draft model on ShareGPT dataset, with token-alignment set to top-$k$ ($k = 3$)

Table 1: Comparison of different speculative decoding methods. This table presents evaluation results on standard LLM benchmarks with temperature $T \in \{0, 1\}$, including speedup ratio $SR$ and acceptance lengths $\tau$. Higher values indicate better performance.

| Model | Method | Temperature = 0 | | | | | | | | Temperature = 1 | | | | | | | |
|---|---|---|---|---|---|---|---|---|---|---|---|---|---|---|---|---|---|
| | | MT-bench | | HumanEval | | GSM8K | | Average | | MT-bench | | HumanEval | | GSM8K | | Average | |
| | | $SR$ | $\tau$ | $SR$ | $\tau$ | $SR$ | $\tau$ | $SR$ | $\tau$ | $SR$ | $\tau$ | $SR$ | $\tau$ | $SR$ | $\tau$ | $SR$ | $\tau$ |
| LLaMA2 Chat 7B | PLD | 1.41 | 1.46 | 1.51 | 1.57 | 1.34 | 1.39 | 1.42 | 1.47 | N/A, since the acceptance conditions are relaxed | | | | | | | |
| | Lookahead | 1.64 | 1.71 | 1.75 | 1.81 | 1.57 | 1.63 | 1.65 | 1.72 | N/A, since the acceptance conditions are relaxed | | | | | | | |
| | EAGLE-2 | 2.69 | 4.50 | 3.22 | 5.24 | 2.77 | 4.72 | 2.89 | 4.82 | 2.41 | 4.29 | 3.00 | 5.01 | 2.63 | 4.66 | 2.68 | 4.65 |
| | FSPAD | 2.89 | 4.82 | 3.38 | 5.62 | 2.95 | 4.99 | 3.07 | 5.14 | 2.61 | 4.53 | 3.14 | 5.35 | 2.84 | 4.88 | 2.86 | 4.92 |
| | HASS | 2.97 | 4.97 | 3.46 | 5.69 | 3.06 | 5.12 | 3.17 | 5.26 | 2.72 | 4.64 | 3.18 | 5.22 | 2.83 | 5.08 | 2.91 | 4.98 |
| | **GRIFFIN** | **3.12** | **5.11** | **3.61** | **5.93** | **3.10** | **5.27** | **3.28** | **5.44** | **2.81** | **4.81** | **3.33** | **5.63** | **3.06** | **5.26** | **3.07** | **5.23** |
| LLaMA3 Instruct 8B | EAGLE | 1.32 | 2.96 | 2.07 | 3.76 | 1.88 | 3.61 | 1.76 | 3.44 | 1.28 | 2.71 | 1.42 | 3.36 | 1.66 | 3.31 | 1.45 | 3.13 |
| | EAGLE-2 | 2.56 | 4.18 | 3.36 | 5.05 | 2.53 | 4.41 | 2.82 | 4.54 | 2.26 | 3.75 | 2.63 | 4.77 | 2.46 | 4.30 | 2.45 | 4.27 |
| | EAGLE-3 | 2.93 | 4.71 | 3.59 | 5.72 | 3.17 | 5.01 | 3.23 | 5.15 | 2.51 | 4.18 | 3.27 | 5.47 | 2.90 | 4.85 | 2.89 | 4.83 |
| | FSPAD | 2.72 | 4.52 | 3.40 | 5.39 | 2.95 | 4.77 | 3.02 | 4.89 | 2.43 | 4.09 | 3.04 | 5.18 | 2.75 | 4.60 | 2.74 | 4.62 |
| | HASS | 2.75 | 4.63 | 3.51 | 5.70 | 3.09 | 5.06 | 3.12 | 5.13 | 2.41 | 4.15 | 3.09 | 5.41 | 2.92 | 4.90 | 2.81 | 4.82 |
| | **GRIFFIN** | **3.09** | **4.85** | **3.65** | **5.97** | **3.30** | **5.31** | **3.35** | **5.38** | **2.62** | **4.35** | **3.31** | **5.62** | **3.07** | **5.08** | **3.00** | **5.02** |
| Vicuna1.5 7B | SPS | 1.81 | 2.34 | 2.04 | 2.68 | 1.73 | 2.28 | 1.86 | 2.43 | 1.49 | 1.85 | 1.57 | 1.99 | 1.52 | 1.80 | 1.53 | 1.88 |
| | Medusa | 1.97 | 2.60 | 2.07 | 2.75 | 1.93 | 2.65 | 1.99 | 2.67 | N/A, since the acceptance conditions are relaxed | | | | | | | |
| | EAGLE-2 | 3.56 | 4.74 | 3.92 | 5.30 | 3.69 | 5.03 | 3.72 | 5.02 | 3.15 | 4.20 | 3.30 | 4.62 | 3.41 | 4.65 | 3.29 | 4.49 |
| | FSPAD | 3.73 | 5.16 | 4.12 | 5.74 | 3.85 | 5.37 | 3.90 | 5.42 | 3.27 | 4.53 | 3.45 | 5.11 | 3.56 | 4.98 | 3.42 | 4.87 |
| | HASS | 3.91 | 5.15 | 4.22 | 5.86 | 3.97 | 5.41 | 4.03 | 5.47 | 3.34 | 4.52 | 3.62 | 5.16 | 3.70 | 5.03 | 3.55 | 4.90 |
| | **GRIFFIN** | **4.02** | **5.36** | **4.53** | **6.29** | **4.14** | **5.63** | **4.23** | **5.76** | **3.38** | **4.64** | **4.12** | **5.68** | **3.88** | **5.29** | **3.79** | **5.20** |
| Qwen2 Instruct 7B | EAGLE-2 | 2.32 | 3.80 | 2.90 | 4.73 | 2.70 | 4.32 | 2.64 | 4.28 | 2.00 | 3.01 | 2.71 | 4.18 | 2.60 | 3.98 | 2.43 | 3.72 |
| | HASS | 2.59 | 4.23 | 3.18 | 5.46 | 2.91 | 4.86 | 2.89 | 4.85 | 2.17 | 3.23 | 2.83 | 4.52 | 2.79 | 4.38 | 2.59 | 4.04 |
| | **GRIFFIN** | **2.76** | **4.67** | **3.34** | **5.75** | **3.02** | **5.13** | **3.04** | **5.18** | **2.27** | **3.36** | **3.04** | **4.82** | **2.96** | **4.71** | **2.76** | **4.30** |
| LLaMA2 Chat 13B | EAGLE-2 | 2.97 | 4.68 | 3.61 | 5.59 | 3.05 | 4.97 | 3.21 | 5.08 | 2.77 | 4.45 | 3.41 | 5.45 | 2.97 | 4.83 | 3.05 | 4.91 |
| | FSPAD | 3.09 | 5.05 | 3.91 | 5.98 | 3.32 | 5.35 | 3.44 | 5.46 | 3.03 | 4.85 | 3.51 | 5.71 | 3.21 | 5.25 | 3.25 | 5.27 |
| | HASS | 3.11 | 5.05 | 4.16 | 6.05 | 3.38 | 5.33 | 3.54 | 5.47 | 3.05 | 4.90 | 3.66 | 5.85 | 3.22 | 5.30 | 3.31 | 5.35 |
| | **GRIFFIN** | **3.33** | **5.27** | **4.29** | **6.26** | **3.61** | **5.56** | **3.74** | **5.70** | **3.36** | **5.07** | **3.94** | **6.13** | **3.61** | **5.49** | **3.64** | **5.56** |
| LLaMA3 Instruct 70B | EAGLE-2 | 2.96 | 4.13 | 4.03 | 5.08 | 3.21 | 4.42 | 3.40 | 4.54 | 3.04 | 4.05 | 3.65 | 5.01 | 3.20 | 4.32 | 3.34 | 4.46 |
| | HASS | 3.36 | 4.59 | 4.61 | 5.73 | 4.01 | 5.21 | 3.99 | 5.17 | 3.35 | 4.48 | 4.23 | 5.65 | 3.84 | 5.17 | 3.80 | 5.10 |
| | **GRIFFIN** | **3.52** | **4.66** | **4.71** | **6.03** | **4.09** | **5.39** | **4.11** | **5.36** | **3.49** | **4.54** | **4.33** | **5.94** | **3.90** | **5.30** | **3.91** | **5.26** |
| Mixtral-v0.1 Instruct 8x7B | EAGLE-2 | 1.96 | 3.39 | 2.34 | 4.13 | 2.19 | 3.79 | 2.16 | 3.77 | 1.93 | 3.32 | 2.28 | 3.98 | 2.09 | 3.71 | 2.10 | 3.67 |
| | HASS | 2.17 | 3.67 | 2.63 | 4.76 | 2.39 | 4.58 | 2.39 | 4.33 | 2.09 | 3.61 | 2.53 | 4.58 | 2.24 | 4.46 | 2.28 | 4.21 |
| | **GRIFFIN** | **2.29** | **3.97** | **2.82** | **5.25** | **2.51** | **4.86** | **2.54** | **4.69** | **2.22** | **3.89** | **2.71** | **5.08** | **2.39** | **4.72** | **2.44** | **4.56** |

for $N = 3$ steps. Other hyperparameters (e.g., optimizer) match EAGLE-2 for fair comparison. We proved the detailed description of GRIFFIN's hyperparameter settings in Appendix. C and implementation of baseline methods in Appendix. D.

We assess performance on three key tasks: multi-turn conversation (MT-Bench [4]), code generation (HumanEval [5]), and mathematical reasoning (GSM8K [14]). To align with prior work (e.g., DistillSpec [35], EAGLE), we fix the batch size to 1 and set the temperature $T \in \{0, 1\}$. Like prior speculative decoding methods, GRIFFIN is also lossless, eliminating the need for additional quality evaluation of generation. Accordingly, we follow priors and focus on acceleration metrics: 1) **Speedup Ratio (SR)** to measure actual test speedup ratio over vanilla autoregressive decoding; and 2) **Acceptance Length** ($\tau$) which is average token number accepted per drafting-verification cycle.

## 5.1 Comparison with SoTAs

We present the acceptance lengths and speedup ratio of various methods across three datasets in Table 1. GRIFFIN consistently achieves the highest acceptance length and speedup ratio across all datasets and LLMs tested. Each GRIFFIN drafting-verification cycle generates approximately 5–6 tokens, significantly exceeding other methods. This is roughly three times the amount of standard speculative sampling and 1.5 times the amount of EAGLE.

For the multi-round conversation task (MT-Bench) with LLaMA3 8B (temperature $T = 0$), GRIFFIN achieves an 8.7% higher speedup ratio compared to HASS. Even for temperature $T = 1$, GRIFFIN

Table 2: Ablation study on Token-Alignable Training (TAT) and Token-Alignable Draft Model (TAD). This table presents the evaluation of speedup ratio $SR$ and acceptance lengths $\tau$ on LLM benchmarks with temperature $T \in \{0, 1\}$. Higher values indicate better performance.

| Method | Temperature = 0 | | | | | | | | Temperature = 1 | | | | | | | |
|---|---|---|---|---|---|---|---|---|---|---|---|---|---|---|---|---|
| | MT-bench | | HumanEval | | GSM8K | | Average | | MT-bench | | HumanEval | | GSM8K | | Average | |
| | $SR$ | $\tau$ | $SR$ | $\tau$ | $SR$ | $\tau$ | $SR$ | $\tau$ | $SR$ | $\tau$ | $SR$ | $\tau$ | $SR$ | $\tau$ | $SR$ | $\tau$ |
| GRIFFIN | **3.12** | **5.11** | **3.61** | **5.93** | **3.10** | **5.27** | **3.28** | **5.44** | **2.81** | **4.81** | **3.33** | **5.63** | **3.06** | **5.26** | **3.07** | **5.23** |
| w/o both | 2.69 | 4.50 | 3.22 | 5.24 | 2.77 | 4.72 | 2.89 | 4.82 | 2.41 | 4.29 | 3.00 | 5.01 | 2.63 | 4.66 | 2.68 | 4.65 |
| w/o TAT | 2.89 | 4.85 | 3.40 | 5.65 | 3.01 | 5.04 | 3.10 | 5.18 | 2.62 | 4.56 | 3.13 | 5.34 | 2.75 | 4.96 | 2.83 | 4.95 |
| w/o TAD | 2.95 | 4.94 | 3.45 | 5.68 | 3.08 | 5.14 | 3.16 | 5.25 | 2.73 | 4.65 | 3.24 | 5.31 | 2.82 | 5.05 | 2.93 | 5.00 |

Table 3: Comparison of different top-$k$ parameter for GRIFFIN. This table presents evaluation of speedup ratio $SR$ and acceptance lengths $\tau$ on standard LLM benchmarks with temperature $T \in \{0, 1\}$. Higher values indicate better performance. NA represents do not align token.

| Top-$k$ | Temperature = 0 | | | | | | | | Temperature = 1 | | | | | | | |
|---|---|---|---|---|---|---|---|---|---|---|---|---|---|---|---|---|
| | MT-bench | | HumanEval | | GSM8K | | Average | | MT-bench | | HumanEval | | GSM8K | | Average | |
| | $SR$ | $\tau$ | $SR$ | $\tau$ | $SR$ | $\tau$ | $SR$ | $\tau$ | $SR$ | $\tau$ | $SR$ | $\tau$ | $SR$ | $\tau$ | $SR$ | $\tau$ |
| 1 | 3.01 | 5.03 | 3.56 | 5.85 | 3.05 | 5.17 | 3.21 | 5.35 | 2.75 | 4.73 | 3.27 | 5.52 | 2.85 | 5.10 | 2.96 | 5.12 |
| 3 | **3.12** | **5.11** | **3.61** | **5.93** | **3.10** | **5.27** | **3.28** | **5.44** | **2.81** | **4.81** | **3.33** | **5.63** | **3.06** | **5.26** | **3.07** | **5.23** |
| 5 | 3.09 | 5.09 | 3.59 | 5.91 | 3.08 | 5.23 | 3.25 | 5.41 | 2.78 | 4.78 | 3.32 | 5.61 | 3.01 | 5.19 | 3.04 | 5.19 |
| 10 | 3.03 | 5.05 | 3.55 | 5.84 | 3.06 | 5.19 | 3.21 | 5.36 | 2.74 | 4.72 | 3.30 | 5.60 | 2.89 | 5.12 | 2.98 | 5.15 |
| NA | 2.97 | 4.95 | 3.52 | 5.76 | 3.04 | 5.12 | 3.18 | 5.28 | 2.72 | 4.65 | 3.24 | 5.47 | 2.82 | 5.01 | 2.93 | 5.04 |

maintains a 7.8% improvement over HASS. For the code generation task (HumanEval) with Vicuna 7B, GRIFFIN demonstrates a 7.3% increase in speedup ratios compared to HASS at a temperature of 0, and a 13.8% improvement when the temperature is set to 1. For the mathematical reasoning task (GSM8K with LLaMA2 13B), GRIFFIN achieves a 6.8% increase in speedup ratios compared to HASS with temperature $T = 0$, and an 12.1% improvement at a temperature of 1. GRIFFIN also outperforms EAGLE-3 across all benchmarks and temperature settings in LLaMA3 8B. GRIFFIN achieves 3.7% higher speedup ratio and 4.5% higher acceptance length compared with EAGLE-3 at a temperature of 0. GRIFFIN achieves 3.8% higher speedup ratio and 3.9% higher acceptance length compared with EAGLE-3 when the temperature is set to 1.

GRIFFIN also demonstrates consistently strong acceleration across LLMs with different architectures beyond LLaMA/Vicuna. On the Qwen2 7B model, GRIFFIN achieves a $5.8\%$ improvement in speedup over HASS. The speedup ratio for Qwen2 7B is slightly lower than LLaMA2 7B model. This discrepancy can be attributed to Qwen2 7B's larger vocabulary size, which results in a more substantial LM Head and subsequently slows down the draft model's decoding speed. For Mixtral-8x7B, the acceleration from speculative decoding, including that of GRIFFIN, is less pronounced compared to other LLMs. This is primarily due to the inherent challenges of applying speculative decoding techniques to Mixture-of-Experts (MoE) architectures. In these settings, verifying multiple tokens simultaneously imposes additional computational overhead, which affects all speculative decoding methods such as GRIFFIN, HASS, and EAGLE-2. Nevertheless, even under the MoE scenario, GRIFFIN achieves more than a $6.6\%$ higher speedup ratio than HASS on Mixtral-8x7B, further demonstrating its strong generalization ability across diverse large language model architectures.

The results across diverse tasks and models highlight the versatility and effectiveness of GRIFFIN. The consistent improvements over HASS, even at different temperatures, underscore GRIFFIN's robustness in handling varying levels of uncertainty in token predictions. Moreover, the performance gains in tasks like code generation and mathematical reasoning suggest that GRIFFIN's token-alignable speculative decoding framework is particularly advantageous for applications requiring high precision and reasoning capabilities. These findings position GRIFFIN as a strong candidate for accelerating LLM inference in real-world scenarios, where both speed and accuracy are critical.

Table 4: Comparision of varied training steps for GRIFFIN. This table presents evaluation results of speedup ratio $SR$ and acceptance lengths $\tau$ on standard LLM benchmarks with temperature $T \in \{0, 1\}$. Higher values indicate better performance.

| Step | Temperature = 0 | | | | | | | | Temperature = 1 | | | | | | | |
|---|---|---|---|---|---|---|---|---|---|---|---|---|---|---|---|---|
| | MT-bench | | HumanEval | | GSM8K | | Average | | MT-bench | | HumanEval | | GSM8K | | Average | |
| | $SR$ | $\tau$ | $SR$ | $\tau$ | $SR$ | $\tau$ | $SR$ | $\tau$ | $SR$ | $\tau$ | $SR$ | $\tau$ | $SR$ | $\tau$ | $SR$ | $\tau$ |
| 1 | 2.89 | 4.85 | 3.40 | 5.65 | 3.01 | 5.04 | 3.10 | 5.18 | 2.62 | 4.56 | 3.13 | 5.34 | 2.75 | 4.96 | 2.83 | 4.95 |
| 2 | 2.99 | 5.02 | 3.51 | 5.81 | 3.06 | 5.15 | 3.19 | 5.33 | 2.74 | 4.73 | 3.26 | 5.49 | 2.92 | 5.14 | 2.97 | 5.12 |
| 3 | 3.12 | 5.11 | 3.61 | 5.93 | 3.10 | 5.27 | 3.28 | 5.44 | 2.81 | 4.81 | 3.33 | 5.63 | 3.06 | 5.26 | 3.07 | 5.23 |
| 4 | 3.13 | 5.13 | 3.62 | 5.96 | 3.12 | 5.31 | 3.29 | 5.47 | 2.82 | 4.84 | 3.35 | 5.66 | 3.08 | 5.31 | 3.08 | 5.27 |
| **5** | **3.13** | **5.14** | **3.63** | **5.98** | **3.13** | **5.33** | **3.30** | **5.48** | **2.83** | **4.86** | **3.36** | **5.68** | **3.10** | **5.34** | **3.10** | **5.29** |

## 5.2 Ablation Study

**Effectiveness of GRIFFIN Components.** We evaluate the impact of GRIFFIN's two key components—Token-Alignable Training (TAT) and the Token-Alignable Draft Model (TAD)—using LLaMA2-Chat 7B. As shown in Table 2, removing either component significantly reduces acceptance length and speed up ratio. Removing TAT leads to a consistent performance drop across all benchmarks, with average acceptance length reduced by 0.26 at $T = 0$ and 0.28 at $T = 1$, speed up ratio reduced by 0.18 at $T = 0$ and 0.24 at $T = 1$. This confirms the importance of TAT in aligning draft tokens during training. Similarly, removing TAD causes noticeable degradation, with acceptance length decreasing by 0.19 at $T = 0$ and 0.23 at $T = 1$, speed up ratio reduced by 0.12 at $T = 0$ and 0.14 at $T = 1$, highlighting TAD's role in reducing misalignment during decoding. Notably, removing both components results in the steepest decline—0.62 at $T = 0$ and 0.58 at $T = 1$—underscoring their complementary effects. Together, TAT and TAD ensure that draft tokens are aligned during both training and decoding, enabling GRIFFIN to achieve state-of-the-art performance.

**Hyper-parameters in Token-Alignable Training.** We analyze the effect of the hyper-parameter $k$ which determines the number of top-$k$ tokens to align. As shown in Table 3, aligning top-$k$ tokens (from 1 to 10) consistently improves acceptance length and speed up ratio compared to no token-alignment. Notably, aligning only the top-1 token is less effective, as it neglects many other tokens that could benefit from alignment. The acceptance length and speed up ratio achieves its peak when $k = 3$, suggesting that aligning a small but sufficient number of tokens provides the optimal trade-off between alignment and generalization.

We further analyze the effect of increasing the number of training steps $N$ in **TAT**. As shown in Table 4, increasing the training steps steadily improves GRIFFIN's acceptance length during the first 5 steps. Unlike HASS which plateaus after step 3 (see Fig. 1 b)), GRIFFIN continues to improve due to its token alignment mechanism. However, as the number of aligned tokens decreases with each additional training step, the improvements become less pronounced at steps 4 and 5. To ensure a fair comparison with HASS, we choose the number of training steps $N = 3$ in our experiments.

Table 5: Ablation study on TGF. This table presents evaluation results of speedup ratio $SR$ and acceptance lengths $\tau$ on standard LLM benchmarks with temperature $T \in \{0, 1\}$. Higher values indicate better performance. "Feature" and "Fused" denotes using $\mathbf{F}$ and $\mathbf{h}$ to replace $\mathbf{x}$ in Eqn. (7).

| Method | Temperature = 0 | | | | | | | | Temperature = 1 | | | | | | | |
|---|---|---|---|---|---|---|---|---|---|---|---|---|---|---|---|---|
| | MT-bench | | HumanEval | | GSM8K | | Average | | MT-bench | | HumanEval | | GSM8K | | Average | |
| | $SR$ | $\tau$ | $SR$ | $\tau$ | $SR$ | $\tau$ | $SR$ | $\tau$ | $SR$ | $\tau$ | $SR$ | $\tau$ | $SR$ | $\tau$ | $SR$ | $\tau$ |
| GRIFFIN | **3.12** | **5.11** | **3.61** | **5.93** | **3.10** | **5.27** | **3.28** | **5.44** | **2.81** | **4.81** | **3.33** | **5.63** | **3.06** | **5.26** | **3.07** | **5.23** |
| Feature | 2.63 | 4.44 | 3.06 | 4.78 | 2.71 | 4.60 | 2.80 | 4.61 | 2.35 | 4.23 | 2.86 | 4.47 | 2.54 | 4.50 | 2.58 | 4.40 |
| Fused | 2.91 | 4.87 | 3.42 | 5.68 | 2.96 | 5.07 | 3.10 | 5.21 | 2.64 | 4.59 | 3.15 | 5.36 | 2.83 | 4.97 | 2.87 | 4.97 |

**Effectiveness of Token-Guided Fusion (TGF).** To assess whether TGF's improvements stem from its token-aware design rather than just increased capacity via using more parameters, we conduct an ablation study in Table 5 by altering the secondary fusion input in Eqn. (7). To investigate this,

we replace token embeddings $\mathbf{x}$ with either raw features $\mathbf{F}$ or the initial fused features $\mathbf{h}$ in Eqn. (7), while keeping model size and training process fixed.

Replacing token embeddings $\mathbf{x}$ with raw features $\mathbf{F}$ in Eqn. (7) reduces acceptance length by 0.83 and speed up ratio by 0.48, indicating that features alone are insufficient to resolve inconsistency. Replacing them with the initial fused features $\mathbf{h}$ in Eqn. (7) performs better—these retain some token information—but still lags behind the original design by 0.26 in acceptance length and 0.2 in speed up ratio. These results confirm that TGF's effectiveness is not due to parameter scaling but stems from the explicit use of token embeddings, which are crucial for correcting inconsistent features and aligning the draft model with the target distribution.

We further investigate the impact of varying the expansion dimension of $\mathbf{W}_u$ in Eqn. (7). As reported in Table 7 of Appendix A.3, decreasing the expansion dimension to 4,096 significantly hampers TGF's capacity to separate essential information from noise, leading to marked reductions in both acceptance length and speedup ratio. Conversely, increasing the expansion dimension to 22,016 results in a slight improvement in acceptance length, attributable to greater representational capacity, but also introduces additional computational overhead, thereby reducing the speedup ratio. These findings validate the expansion dimension choice in GRIFFIN, demonstrating a well-balanced trade-off between performance and computational efficiency.

# 6 Conclusion

In this paper, we present GRIFFIN, a token-alignable speculative decoding framework. Prior methods have largely ignored the token misalignment problem between training and decoding. GRIFFIN addresses this by introducing a token-alignable training strategy that excludes misaligned tokens from loss computation. It further incorporates a token-alignable draft model that substantially reduces misalignment. Extensive evaluations across diverse LLMs and datasets show that GRIFFIN consistently outperforms SoTAs, achieving the highest speedup ratios and acceptance lengths.

**Limitations.** GRIFFIN adopts a multi-step training process for token-alignable training, which incurs additional training overhead compared to EAGLE. However, since the draft model is trained only once, real-world applications prioritize decoding efficiency over training overhead, as inference is the primary bottleneck. GRIFFIN improves the speedup ratio by over 18% compared to EAGLE2, making the extra training cost a worthwhile trade-off for the significant inference acceleration it delivers. Furthermore, GRIFFIN's overall training overhead remains comparable to that of HASS. Under the same training cost, GRIFFIN achieves an over 7% improvement in speedup ratio compared to HASS, further highlighting its effectiveness.

**Broader Impact.** GRIFFIN advances the efficiency of LLM inference by accelerating decoding speed without sacrificing output quality. This improvement can democratize access to powerful LLMs by making real-time applications more feasible. Downstream, GRIFFIN could enable smoother, faster interactive AI for education, healthcare assistants, accessibility tools, and scientific research, broadening beneficial applications and reducing latency barriers for users worldwide.

# Acknowledgement

This work was supported by the Yangtze River Delta Science and Technology Innovation Community Joint Research Project (YDZX20233100004031), the National Key Research and Development Program of China (2022YFC3302300), and the Singapore Ministry of Education (MOE) Academic Research Fund (AcRF) Tier 1 grant (Proposal ID: 25-SIS-SMU-003). Any opinions, findings and conclusions or recommendations expressed in this material are those of the author(s) and do not reflect the views of the Ministry of Education, Singapore.

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

# A    Analysis for the Architecture of Token-Guided Fusion (TGF)

## A.1    Motivation Behind TGF

The Token-Guided Fusion (TGF) module is motivated by the limitations of the standard concat-then-MLP strategy, as adopted in EAGLE, which does not fully capture the complementary information between token embeddings and draft model features. In practice, features generated by the draft model often remain misaligned with the target model's representations, a discrepancy that cannot be effectively eliminated with feature-level loss minimization alone. TGF addresses this challenge by explicitly leveraging token embeddings to guide the fusion process, aligning feature distributions more closely to those of the target model. As confirmed by ablation results (Table 5), this targeted architectural enhancement significantly reduces feature inconsistency, demonstrating that the modest complexity introduced by TGF provides strong empirical gains.

**Key Architectural Enhancements in TGF:**

- **Feature Normalization and Dimensional Expansion:** Separate layer normalization is applied to both the initial fused features $\mathbf{h}$ and token embeddings $\mathbf{x}$ in Fig. 3 (b-ii), allowing for independent statistical scaling and improved stability during training. The Up Projector in Fig. 3 (b-ii) expands the feature dimensionality, which helps disentangle relevant information and increase the expressiveness of fused representations.
- **Nonlinear Refinement and Consolidation:** The SiLU activation function in Fig. 3 (b-iii) introduces nonlinearity, enhancing the module's capacity to recover complex feature interactions beyond linear operations. Afterwards, the Down Projector in Fig. 3 (b-iii) compresses the representation back to the target dimension, aggregating enriched information into a stable fused feature for downstream processing.

Overall, TGF enables the draft model to more accurately approximate the target model's output space, as evidenced by measurable improvements in acceptance lengths and speedup ratio as shown in Table. 1.

## A.2    Ablation Study on the TGF Architecture

To systematically assess the contributions of each component within the TGF module, we performed targeted ablation experiments, with each variant constructed by selectively removing or modifying specific submodules:

- **Ablation 1:** Simultaneous removal of the Up Projector (Fig. 3 (b-ii)) and the SiLU activation (Fig. 3 (b-iii)).
- **Ablation 2:** Exclusion of the token embeddings $\mathbf{x}$ from the secondary fusion step in Eqn. (7).
- **Ablation 3:** Exclusion of the initial fused feature $\mathbf{h}$ from the secondary fusion step in Eqn. (7).

Table 6: Ablation results for architecture of TGF. This table presents evaluation results on standard LLM benchmarks with temperature $T \in \{0, 1\}$, including speedup ratio $SR$ and acceptance lengths $\tau$. Higher values indicate better performance.

| Method | Temperature = 0 | | | | | | | | Temperature = 1 | | | | | | | |
|---|---|---|---|---|---|---|---|---|---|---|---|---|---|---|---|---|
| | MT-bench | | HumanEval | | GSM8K | | Mean | | MT-bench | | HumanEval | | GSM8K | | Mean | |
| | $SR$ | $\tau$ | $SR$ | $\tau$ | $SR$ | $\tau$ | $SR$ | $\tau$ | $SR$ | $\tau$ | $SR$ | $\tau$ | $SR$ | $\tau$ | $SR$ | $\tau$ |
| GRIFFIN | **3.12** | **5.11** | **3.61** | **5.93** | **3.10** | **5.27** | **3.28** | **5.44** | **2.81** | **4.81** | **3.33** | **5.63** | **3.06** | **5.26** | **3.07** | **5.23** |
| Ablation 1 | 2.93 | 4.85 | 3.36 | 5.73 | 3.04 | 5.07 | 3.11 | 5.21 | 2.68 | 4.55 | 3.20 | 5.39 | 2.69 | 4.96 | 2.86 | 4.96 |
| Ablation 2 | 3.02 | 4.97 | 3.51 | 5.83 | 3.10 | 5.22 | 3.21 | 5.34 | 2.73 | 4.63 | 3.31 | 5.58 | 2.92 | 5.09 | 2.99 | 5.10 |
| Ablation 3 | 1.76 | 2.75 | 1.92 | 3.09 | 1.65 | 2.58 | 1.78 | 2.81 | 1.67 | 2.65 | 1.88 | 3.04 | 1.49 | 2.53 | 1.68 | 2.74 |

Table 6 presents the results of these ablation settings. The following key observations can be drawn:

- **Ablation 1:** Omitting both the Up Projector and SiLU activation produces a marked decrease in performance, with acceptance length reduced by 0.23 ($T = 0$) and 0.27 ($T = 1$), and speedup ratio reduced by 0.17 ($T = 0$) and 0.21 ($T = 1$). This highlights the critical role these components play in enabling expressive and stable feature fusion.

- **Ablation 2:** Removing token embeddings $\mathbf{x}$ from the secondary fusion step adversely affects the model's ability to inject token-specific information, resulting in lower acceptance length (by 0.10 at $T = 0$, 0.13 at $T = 1$) and speedup ratio (by 0.07 at $T = 0$, 0.08 at $T = 1$).

- **Ablation 3:** Excluding the initially fused feature $\mathbf{h}$ from the secondary fusion produces the most severe degradation: acceptance length decreases by 2.63 ($T = 0$) and 2.49 ($T = 1$), while speedup ratio drops by 1.50 ($T = 0$) and 1.39 ($T = 1$). This underscores that the recurrent integration of fused features is indispensable for capturing high-quality representations and achieving effective alignment.

Overall, these ablation results confirm the necessity of each architectural component within TGF for maximizing acceptance length and speed up ratio.

### A.3 Ablation on the Expansion Dimension of TGF

In the TGF module, the expansion dimension refers to the output dimensionality of the Up Projector in Fig. 3 (b-ii). For GRIFFIN, we set this dimension to 11,008, matching the intermediate size of the target model's feed-forward network (FFN). To evaluate the impact of this design choice, we perform ablation experiments by varying the expansion dimension, while holding all other components and training protocols constant.

Table 7: Comparison of different expansion dimension of GRIFFIN. This table presents evaluation results on standard LLM benchmarks with temperature $T \in \{0, 1\}$, including speedup ratio $SR$ and acceptance lengths $\tau$. Higher values indicate better performance.

| | | Temperature = 0 | | | | | | | | Temperature = 1 | | | | | | | |
|---|---|---|---|---|---|---|---|---|---|---|---|---|---|---|---|---|---|
| | | MT-bench | | HumanEval | | GSM8K | | Mean | | MT-bench | | HumanEval | | GSM8K | | Mean | |
| Expansion Dimension | Draft Model Size | SR | $\tau$ | SR | $\tau$ | SR | $\tau$ | SR | $\tau$ | SR | $\tau$ | SR | $\tau$ | SR | $\tau$ | SR | $\tau$ |
| 11,008 (GRIFFIN) | 0.41B | **3.12** | 5.11 | **3.61** | 5.93 | **3.10** | 5.27 | **3.28** | 5.44 | **2.81** | 4.81 | **3.33** | 5.63 | **3.06** | **5.26** | **3.07** | 5.23 |
| 4,096 | 0.33B | 3.06 | 5.02 | 3.55 | 5.82 | 3.06 | 5.13 | 3.22 | 5.32 | 2.75 | 4.68 | 3.27 | 5.51 | 3.02 | 5.11 | 3.01 | 5.10 |
| 22,016 | 0.55B | 2.97 | **5.15** | 3.38 | **6.01** | 2.95 | **5.34** | 3.10 | **5.50** | 2.60 | **4.84** | 3.14 | **5.76** | 2.71 | 5.17 | 2.82 | **5.25** |

Table 7 summarizes the results, from which we draw the following conclusions:

- **Smaller expansion (4,096):** Lowering the expansion dimension to 4,096 degrades TGF's capacity to extract and distinguish salient features, leading to a notable reduction in acceptance length (by 0.12 at $T = 0$, 0.13 at $T = 1$) and speedup ratio (by 0.06 at both $T = 0$ and $T = 1$).

- **Larger expansion (22,016):** Increasing the expansion dimension to 22,016 yields a slight improvement in acceptance length (by 0.06 at $T = 0$, 0.02 at $T = 1$), suggesting marginal gains in representational power. However, this is offset by a decline in speedup ratio (reduced by 0.18 at $T = 0$, 0.25 at $T = 1$), primarily due to increased computational overhead and an additional 0.14B parameters.

Overall, these results validate our chosen configuration: setting the TGF expansion dimension equal to the target model's FFN intermediate size achieves an effective balance between fusion performance and computational efficiency.

## B Effectiveness of Token-Alignable Draft Model (TAD) Components

We assess the individual contributions of Token-Guided Fusion (TGF) and Token-Enhanced Head (TEH)—the two principal components of the Token-Alignable Draft model (TAD)—using LLaMA2-Chat 7B as the base model. Ablation results are summarized in Table 8.

Table 8: Ablation results for TAD. This table presents evaluation results on standard LLM benchmarks with temperature $T \in \{0, 1\}$, including speedup ratio $SR$ and acceptance lengths $\tau$. Higher values indicate better performance.

| Method | Temperature = 0 | | | | | | | | Temperature = 1 | | | | | | | |
|---|---|---|---|---|---|---|---|---|---|---|---|---|---|---|---|---|
| | MT-bench | | HumanEval | | GSM8K | | Mean | | MT-bench | | HumanEval | | GSM8K | | Mean | |
| | $SR$ | $\tau$ | $SR$ | $\tau$ | $SR$ | $\tau$ | $SR$ | $\tau$ | $SR$ | $\tau$ | $SR$ | $\tau$ | $SR$ | $\tau$ | $SR$ | $\tau$ |
| GRIFFIN | **3.12** | **5.11** | **3.61** | **5.93** | **3.10** | **5.27** | **3.28** | **5.44** | **2.81** | **4.81** | **3.33** | **5.63** | **3.06** | **5.26** | **3.07** | **5.23** |
| w/o TEH | 3.09 | 5.07 | 3.56 | 5.85 | 3.08 | 5.22 | 3.24 | 5.38 | 2.77 | 4.75 | 3.29 | 5.55 | 3.03 | 5.21 | 3.03 | 5.17 |
| w/o TGF | 3.04 | 4.99 | 3.49 | 5.75 | 3.05 | 5.13 | 3.19 | 5.29 | 2.75 | 4.70 | 3.25 | 5.43 | 2.88 | 5.11 | 2.96 | 5.08 |

Removing either component leads to a clear and consistent reduction in both acceptance length and speedup ratio:

- **Token-Enhanced Head (TEH):** Excluding TEH results in a consistent performance drop across all benchmarks, with the average acceptance length reduced by 0.06 and speedup ratio decreased by 0.04. This highlights the critical role of TEH in boosting the draft model's token prediction accuracy.

- **Token-Guided Fusion (TGF):** Excluding TGF leads to even greater degradation: acceptance length drops by 0.15, and speedup ratio decreases by 0.09 at $T = 0$ and by 0.11 at $T = 1$. These findings reinforce TGF's efficacy in mitigating feature misalignment during speculative decoding.

Collectively, these results underscore that both TGF and TEH are indispensable for maximizing the effectiveness and efficiency of the TAD architecture.

## C   Implementation Details of GRIFFIN

### C.1   Loss function

**Per-step loss composition.**   The per-step loss $\ell(\bar{\mathbf{x}}_t, \mathbf{x}_t, \bar{\mathbf{F}}_t, \mathbf{F}_t)$ in Eq. (5) combines two complementary components that supervise both the token prediction and the latent feature alignment of the draft model $\mathcal{M}$ with the target model $\mathcal{T}$:

$$\ell(\bar{\mathbf{x}}_t, \mathbf{x}_t, \bar{\mathbf{F}}_t, \mathbf{F}_t) = \lambda_{\text{tok}}\, \ell_{\text{tok}}(\bar{\mathbf{x}}_t, \mathbf{x}_t) + \lambda_{\text{feat}}\, \ell_{\text{feat}}(\bar{\mathbf{F}}_t, \mathbf{F}_t), \tag{9}$$

where $\lambda_{\text{tok}}$ and $\lambda_{\text{feat}}$ are scalar weights (default: $\lambda_{\text{tok}} = 1$, $\lambda_{\text{feat}} = 0.1$ unless otherwise stated).

**Token-level loss.**   The token-level supervision aligns the predicted token distribution of the draft model with the ground truth:

$$\ell_{\text{tok}}(\bar{\mathbf{x}}_t, \mathbf{x}_t) = -\log P_{\mathcal{M}}(\mathbf{x}_t \mid \mathbf{x}_{1:t-1}), \tag{10}$$

which corresponds to the standard cross-entropy between the predicted logits $\bar{\mathbf{x}}_t$ and the one-hot target $\mathbf{x}_t$.

**Feature-level loss.**   To encourage the internal representations of the draft and target models to remain consistent, we minimize the $\ell_1$ distance between their hidden features:

$$\ell_{\text{feat}}(\bar{\mathbf{F}}_t, \mathbf{F}_t) = \|\bar{\mathbf{F}}_t - \mathbf{F}_t\|_1. \tag{11}$$

This term regularizes the draft model toward the target model's latent space, facilitating stable alignment across multi-pass decoding.

### C.2   Draft Tree Structure

For all experiments, we use a dynamic tree structure with a total of 60 draft tokens and set the draft tree depth to 6, closely following the optimal configuration established in EAGLE-2 and HASS.

### C.3 Training Configuration

The draft model is trained using the AdamW optimizer, with the following key settings:

- **Learning rate:** $3\mathrm{e}^{-5}$
- **Batch size:** 4 (per GPU)
- **Number of epochs:** 20
- **Total training steps:** 800,000
- **Warmup:** 2,000 steps of linear warmup; learning rate scheduler enabled
- **Optimizer:** AdamW, with betas $(0.9, 0.95)$
- **Gradient clipping:** 0.5 (by value)
- **Maximum sequence length:** 2,048 tokens

All hyperparameters are kept fixed for all reported experiments unless otherwise specified. Additional hyperparameters and implementation scripts are provided in https://github.com/hsj576/GRIFFIN.

## D Clarification of Baseline Methods

For EAGLE, EAGLE-2, and Medusa, we directly utilized the publicly released draft model parameters provided by the respective authors. For methods that do not require draft model training, such as PLD, Lookahead, and SPS, we evaluated performance using official code from their GitHub repositories.

Regarding HASS, we used publicly released draft model parameters for LLaMA2-7B, LLaMA3-8B, LLaMA2-13B, and LLaMA3-70B. However, at the time of submission, official draft model parameters for Vicuna-7B, Qwen2-7B, and Mixtral-8x7B were unavailable. To enable fair comparison, we trained the HASS draft models ourselves using their official GitHub repository and strictly followed the configurations described in the HASS paper. The experimental results we obtained closely corresponded to those reported by the HASS paper. Similarly, draft model parameters for FSPAD were not publicly available at the time of submission. We therefore trained FSPAD's draft models with their official code and under the settings specified in the FSPAD paper. Our experimental outcomes showed high consistency with the results published by the original FSPAD paper.

Regarding EAGLE-3, since EAGLE-3 doesn't provide pre-trained draft models for LLaMA3-8B, we used their official code to train on the ShareGPT dataset, maintaining all other hyperparameters consistent with their paper. Training EAGLE-3 on ShareGPT alone required over 300 A100-80G GPU hours. Following their paper's full protocol (UltraChat-200K + ShareGPT) would require approximately 2,400 GPU hours, which exceeded our computing resources. However, both EAGLE-3 and GRIFFIN used identical ShareGPT training data, ensuring fair comparison.

To ensure the validity of our comparisons, we aligned all key training settings, including dataset, optimizer, and hyperparameters, with those used by EAGLE-2 and HASS. For example, we matched the training procedure to HASS's three-step schedule, ensuring consistency and reliability across all experiments.

## E Parameter Sizes of GRIFFIN's Draft Models

For 7B, 8B, 13B, and 70B scale target models, the corresponding GRIFFIN draft model sizes are 0.41B, 0.42B, 0.65B, and 2.07B parameters, respectively. For Mixtral-8x7B, the draft model size is 0.45B parameters.

By comparison, the draft model sizes for EAGLE-2 and HASS are 0.24B, 0.25B, 0.37B, and 0.99B, while those for FSPAD are 0.42B, 0.43B, 0.67B, and 2.09B, across corresponding target models. For Mixtral-8x7B, the EAGLE-2 and HASS draft model size is 0.28B. Therefore, GRIFFIN's draft model contains between 0.17B and 1.08B more parameters than those of EAGLE-2 and HASS, but remains similar in size to FSPAD's.

Despite this modest increase in parameters, GRIFFIN consistently achieves an average speedup improvement exceeding 8%, as shown in Table 1. The additional parameter count incurs only

marginal computational overhead, which is amply justified by the significant gains in inference efficiency and overall performance.

# F   Training Overhead of GRIFFIN

All the methods(GRIFFIN, EAGLE-2, FSPAD, and HASS) utilize the ShareGPT dataset for draft model training, ensuring an equal number of training tokens across methods.

In terms of computational resources, GRIFFIN employs the same multi-stage training strategy as the state-of-the-art HASS method, with both adopting a three-step training regimen. For 7B, 13B, and 70B parameter models, HASS typically requires approximately 130, 220, and 500 NVIDIA A100 80G GPU hours, respectively, whereas GRIFFIN's requirements are about 150, 250, and 600 NVIDIA A100 80G GPU hours.

Crucially, the draft model is trained only once but leveraged extensively during inference. Thus, in practical scenarios, the primary computational cost lies in the decoding phase. GRIFFIN offers roughly an 8% improvement in speculative decoding speed over HASS, meaning that the slight increase in training overhead is well justified by the substantial gains in inference efficiency.

# G   Throughput of GRIFFIN

To evaluate GRIFFIN's performance under batch sizes greater than 1, we integrated it into the open-source **vLLM** framework, following the same speculative decoding interface used by **EAGLE**. All experiments were conducted on the **LLaMA3-8B-Instruct** model using the **MT-Bench** dataset, with a decoding temperature of 0. We report throughput (tokens per second) relative to the baseline vLLM decoding without any speculative methods.

**Results.**   Table 9 summarizes relative speedups across different batch sizes. GRIFFIN consistently achieves higher throughput than both EAGLE and HASS for all evaluated batch configurations.

Table 9:   Throughput comparison under different batch sizes. Numbers denote relative speedup ($\times$) over vanilla vLLM decoding (1.00$\times$ baseline). All speculative methods were evaluated using sequential speculation with a maximum chain length of 2.

| Batch Size | 2 | 4 | 8 | 16 |
|---|---|---|---|---|
| EAGLE | 1.37× | 1.32× | 1.28× | 1.18× |
| HASS | 1.40× | 1.35× | 1.30× | 1.20× |
| **GRIFFIN** | **1.52×** | **1.45×** | **1.37×** | **1.25×** |

**Implementation constraints.**   These evaluations were conducted under certain restrictions imposed by the current speculative decoding support in vLLM. Specifically, the implementation does **not support tree-based drafting**, which is a key component of our full decoding algorithm. Consequently, all measurements used sequential speculation with a maximum chain length of 2. Therefore, the throughput values in Table 9 are not directly comparable to the main-text results, which were obtained using our native decoding backend configured for tree-structured speculation.

**Analysis.**   The observed trend of decreasing relative speedup as batch size increases is expected and consistent with theoretical expectations. Larger batch sizes improve GPU utilization for the target model, reducing redundant computations and narrowing the efficiency gap between speculative and standard decoding. Moreover, as batches grow, the memory footprint and compute overhead associated with additional draft-model evaluations become increasingly significant, diminishing net throughput gains.

**Discussion.**   Despite these challenges, GRIFFIN maintains substantial advantages—achieving **6–11% higher throughput than EAGLE** and **4–8% higher than HASS** across the tested batch sizes. These improvements demonstrate that GRIFFIN's alignment mechanism continues to yield benefits even in large-batch, high-throughput inference regimes that are typical in production deployments.

## H  Breakdown of Decoding Latency

**Motivation.**   While speculative decoding yields substantial efficiency gains, the overall speedup is bounded by the additional computation required by the draft model. In Table 1 of the main text, the observed *speedup ratio* (SR) is notably lower than the corresponding *acceptance length*. This discrepancy arises primarily from the non-negligible latency overhead of draft model inference.

**Latency formulation.**   Let $N$ denote the total number of tokens generated during decoding. For standard autoregressive decoding, the total decoding latency is

$$\mathbf{T}_a = N \cdot \mathbf{t}, \tag{12}$$

where $\mathbf{t}$ is the average per-token latency of a single forward pass through the target model.

For speculative decoding, at each cycle the target model verifies $\tau$ candidate tokens produced by the draft model with rollout depth $\mathbf{d}$. The corresponding total latency can be approximated as

$$\mathbf{T}_s = \frac{N}{\tau} \cdot \left(\mathbf{t} + \mathbf{d} \cdot \bar{\mathbf{t}}\right), \tag{13}$$

where $\bar{\mathbf{t}}$ is the draft model's average per-pass latency. The resulting theoretical speedup ratio therefore becomes:

$$\text{Speedup Ratio (SR)} = \frac{\mathbf{T}_a}{\mathbf{T}_s} = \frac{\mathbf{t}}{\mathbf{t} + \mathbf{d} \cdot \bar{\mathbf{t}}} \cdot \tau. \tag{14}$$

**Empirical estimation.**   Using the **LLaMA3-8B-Instruct** model on an **A100-80G** GPU as a representative setup, we measure the forward-pass latency of the target model as approximately $\mathbf{t} = 25\,\text{ms}$ and of the draft model as $\bar{\mathbf{t}} = 1.5\,\text{ms}$. If we set the acceptance length to $\tau = 5$ and draft rollout depth $\mathbf{d} = 6$, Eq. (14) gives:

$$\text{SR} = \frac{25}{25 + 6 \times 1.5} \times 5 = 3.68\times,$$

which closely matches our empirical results. This quantitative agreement confirms that the latency contributed by draft model inference is the primary factor limiting the achievable speedup.

**Discussion.**   Although draft model latency constitutes a relatively small portion of the total budget, its accumulation over multiple rollout steps can substantially reduce overall efficiency, particularly for deep or large-$\mathbf{d}$ speculative configurations. Future efforts will explore techniques to further mitigate this cost, such as:

- **Draft-model distillation** to reduce forward-pass complexity;
- **Asynchronous drafting** that overlaps draft and target evaluations;
- **Kernel fusion and caching** to minimize memory transfer overhead.

These analyses confirm that the gap between acceptance length and speedup ratio is quantitatively explained by draft inference latency, and they motivate further system-level optimizations.

## I  Discussion with EAGLE-3

**Motivation.**   EAGLE-3 recently proposed a simplified speculative decoding framework that removes the feature-prediction loss from the draft model objective. To examine the practical effect of this choice and its interaction with our token-alignment mechanisms, we conducted two complementary studies: (i) an *ablation* of GRIFFIN in which the feature-level loss term was removed, and (ii) a *head-to-head comparison* between our implementation of EAGLE-3 and full GRIFFIN under matched training conditions.

**Experimental setup.**   All experiments were performed on **LLaMA3-8B-Instruct** using three standard evaluation suites—**MT-Bench**, **HumanEval**, and **GSM8K**—at decoding temperatures $t{=}0$ and $t{=}1$. For the EAGLE-3 baseline, we trained a draft model following their official open-source repository and hyperparameter settings, including identical optimizer, learning-rate schedule, and architecture. Due to computational constraints, training used the **ShareGPT** dataset only (excluding the additional UltraChat-200K corpus), which would otherwise require roughly 2,400 GPU hours to reproduce fully.

**Results.** Table 10 reports speedup ratio ($SR$) and acceptance length ($\tau$) across the three benchmarks and two temperature settings.

Table 10: Comparison between GRIFFIN, its feature-loss ablation, and EAGLE-3 on **LLaMA3-8B-Instruct**. This table reports results on standard LLM benchmarks (**MT-Bench**, **HumanEval**, **GSM8K**) for temperatures $T \in \{0, 1\}$, including speedup ratio $SR$ and acceptance length $\tau$. Higher values indicate better performance.

| Method | Temperature = 0 | | | | | | | | Temperature = 1 | | | | | | | |
| --- | --- | --- | --- | --- | --- | --- | --- | --- | --- | --- | --- | --- | --- | --- | --- | --- |
| | MT-Bench | | HumanEval | | GSM8K | | Mean | | MT-Bench | | HumanEval | | GSM8K | | Mean | |
| | $SR$ | $\tau$ | $SR$ | $\tau$ | $SR$ | $\tau$ | $SR$ | $\tau$ | $SR$ | $\tau$ | $SR$ | $\tau$ | $SR$ | $\tau$ | $SR$ | $\tau$ |
| **GRIFFIN** | **3.09** | **4.85** | **3.65** | **5.97** | **3.30** | **5.31** | **3.35** | **5.38** | **2.62** | **4.35** | **3.31** | **5.62** | **3.07** | **5.08** | **3.00** | **5.02** |
| w/o FeatLoss | 2.61 | 4.33 | 3.32 | 5.15 | 2.76 | 4.58 | 2.89 | 4.69 | 2.32 | 4.02 | 2.85 | 4.97 | 2.49 | 4.36 | 2.55 | 4.45 |
| EAGLE-3 | 2.93 | 4.71 | 3.59 | 5.72 | 3.17 | 5.01 | 3.23 | 5.15 | 2.51 | 4.18 | 3.27 | 5.47 | 2.90 | 4.85 | 2.89 | 4.83 |

**Observation 1: Importance of feature-level loss.** Removing the feature-prediction term from GRIFFIN produces a consistent degradation of roughly 10–15% in both $SR$ and $\tau$ across all evaluation settings. This confirms that the feature-level supervision remains critical for stabilizing token alignment by enforcing coherence between the hidden representations of the draft and target models.

**Observation 2: Comparison with EAGLE-3.** Under identical training data and inference settings, full GRIFFIN outperforms EAGLE-3 on all metrics. This indicates that EAGLE-3's removal of the feature loss does not yield an advantage at this data scale and that GRIFFIN's alignment mechanisms lead to more efficient speculative decoding even when controlling for model and data size.

**Observation 3: Applicability of GRIFFIN techniques.** EAGLE-3 does not explicitly address token misalignment during training or decoding. In contrast, GRIFFIN introduces the *Token-Alignable Draft* (TAD) architecture and the *Token-Alignable Training* (TAT) procedure, both designed to mitigate this issue. Importantly, these techniques are modular and could in principle be applied to EAGLE-3-style draft models without altering their external decoding interface, potentially improving alignment and throughput performance.

**Summary.** This study demonstrates that feature-level supervision remains beneficial even when the draft model is trained with large-scale data, and that GRIFFIN's token-alignment strategy provides complementary improvements beyond what EAGLE-3 achieves. We hope these findings clarify the design impact of feature prediction loss and encourage future integration of token-alignment principles into other speculative decoding frameworks.

