# OpenReview forum: "GRIFFIN: Effective Token Alignment for Faster Speculative Decoding"
_NeurIPS.cc/2025/Conference — NeurIPS 2025 poster_

### Official Review · Reviewer_vyph · 2025-06-04

**Clarity:** 2
**Significance:** 3
**Originality:** 3
**Rating:** 4
**Confidence:** 3

**Summary:**

GRIFFIN is a speculative decoding framework that enhances the efficiency of large language model inference by explicitly addressing token misalignment and feature misalignment, overlooked issues in prior work. It introduces a token-alignable training strategy that uses dynamic loss masking to exclude misaligned tokens during optimization, and two extra modules: TGF and TEH.

**Questions:**

1. During speculative decoding, all drafted token after misaligned token should be rejected right? This means that actually there is no point to correct the following tokens after a misaligned token. So the goal of token alignment is actually drop "hard" prediction right? If the ground truth is not in top-k of draft model's prediction, then this loss is not counted as this prediction is too hard for the draft model. In this sense, the goal of "token alignment" is totally different from "feature alignment", where "feature alignment" is using draft-model-generated features, indeed aligning train input distribution with test input distribution. Please correct me if my understanding is wrong. If my understanding is right, then I think it lacks a clear explanation and analysis about this in the article. I think it even shouldn't be called "token misalignment", as this mask doesn't actually "align" the token.
2. I didn't understand about why "we gradually shift the model toward using its own outputs during training" in Line 139 and "By progressively adapting draft model to operate under its own predictions and aligning its training with decoding" in Line 180. How "gradually" and "progressively" work? Doesn't in Equation 4, the target-model features is directly replaced with draft-model-generated features?
3. I'm confused about the term: step and pass. In Line 141, "where each training step n involves draft model performing n forward passes to predict n future tokens and their corresponding features.", is this "n" in "each training step n" an index or a total number? And I don't understand why acceptance lengths can be longer than Step in Table 4. Doesn't "step" represent the maximum length of drafted tokens be predicted within a cycle?
4. What is the definition of loss in Equation 2? How did it combine the l1 distance between feature and cross entropy between tokens. If there is a weighted factor, what is the weighted factor? Why this is not included in hyperparameters?
5. The Speedup Ratio is drafting + verification or just drafting? How many GPU does verification used for parallel verifying? Why Line 221 says "For consistency, all inference runs use one NVIDIA A100 80G GPU"? If drafting and verificaiton are on same GPU, speculative decoding should be slower than autogressive decoding, as FLOPs increased.

**Ethical Concerns:**

["NO or VERY MINOR ethics concerns only"]

**Final Justification:**

Authors solved my concerns and I raised my score to positive.

**Limitations:**

Limitation is addressed in section 6.

**Paper Formatting Concerns:**

NaN

**Quality:**

3

**Strengths And Weaknesses:**

Strengths:
1. The experiments are thorough, in which GRIFFIN shows large improvements upon other baselines.
2. The motivation is reasonable in concept.

Weakness:
1. Please check the question.

---

> ### Author Rebuttal · Authors · 2025-07-29
>
> Thank you for your thoughtful feedback. Below, we provide a point-by-point response to address your concerns.
>
> **1. Goal of Token Alignment**. There are some misunderstandings. We would like to clarify the distinctions and objectives of our method.
>
> **Feature Alignment vs Token Alignment.** As illustrated in Fig. 2, **feature misalignment** occurs because during decoding, the draft model uses draft features $ \bar{\mathbf{F}} _ t $ to generate $ \bar{\mathbf{F}} _ {t+1} $, while during training it uses target features $ \mathbf{F} _ t $. **Token misalignment** arises because during decoding, the draft model uses draft tokens $ \bar{\mathbf{x}} _ t $ to generate $ \bar{\mathbf{F}} _ {t+1} $, while during training it uses ground truth tokens $ \mathbf{x} _ t $.
>
> Feature alignment addresses feature misalignment by replacing $ \mathbf{F} _t$ with $ \bar{\mathbf{F}} _t$ during training. However, token misalignment cannot be directly resolved by simply replacing $ \mathbf{x} _t$ with $ \bar{\mathbf{x}} _t$ because we follow the fixed-dataset setting adopted by EAGLE and HASS. We provide the detailed explanations for this constraint in Section 3, lines 124-131. Therefore, GRIFFIN simultaneously incorporates a **token-alignable training strategy** and a **token-alignable draft model** to mitigate misalignment. The training strategy employs a loss masking mechanism to exclude highly misaligned tokens during training, preventing them from negatively impacting the draft model's optimization. The token-alignable draft model introduces input tokens to correct inconsistencies in generated features.
>
> **Why Token Alignment ≠ "Dropping Hard Predictions".**
> The goal of our masking is **not** to discard tokens whose ground-truth falls outside the top-$k$ of the draft model. Here we take one example for illustration. Assume the draft model have generated five draft tokens $ \bar{\mathbf x} _{t+1:t+5}$. If $ \mathbf x _{t+1}$ and $ \mathbf x _{t+2}$ are in top-$ k(\bar{\mathbf{x}} _{t+1})$ and top-$ k(\bar{\mathbf{x}} _{t+2})$ respectively, but $ \mathbf{x} _{t+3}$ is not, we still compute the loss for **all three** positions $ \bar{\mathbf x} _{t+1:t+3}$. Tokens **after** the first mismatch ($ \bar{\mathbf x} _{t+4:t+5}$) are guaranteed to be rejected in verification; training on them provides no useful signal. These positions are therefore **masked**, not because they are “hard,” but because they are **inevitably discarded** during speculative decoding.
>
> **Corrections and Clarifications.** We also checked the whole work to find the reason which leads to your misunderstanding. Firstly, in line 158, we use "aligned" to indicate when the ground-truth token $ \mathbf{x} _t$ appears within top-$k$($ \bar{\mathbf{x}} _t$). We agree that "predictable" would be more accurate than "aligned" in this context, since this "aligned" does not have the same meaning with "token alignment" which refers to the consistency between training and inference in terms of token level. Secondly, Eqn (3) has an incorrect superscript, and should be $ \mathbf{m}_t = \prod _{i=t-n+1}^{t-1}\bar{\mathbf{m}} _i$. This means we do not directly mask out the loss for draft tokens where $ \bar{\mathbf{m}} _{t}=0$.  We will correct them to enhance clarity and readability.
>
> **2. How "gradually" and "progressively" work?**
> Assume the draft model aims to predict $N$ draft tokens $ \bar{\mathbf x} _{t+1:t+N}$ by performing forward pass $N$ times:
>
> 1. **Forward pass 1 (predict $ \bar{\mathbf x} _{t+1}$)**
>
>    • Inputs: ground-truth tokens $ \mathbf{x} _{1:t}$ and target-model features $ \mathbf{F} _{1:t}$.
>
>    • Outputs: token $ \bar{\mathbf x} _{t+1}$ and feature $ \bar{\mathbf F} _{t+1}$.
>
> 2. **Forward pass $n$ (predict $ \bar{\mathbf x} _{t+n}$), $2 ≤ n ≤ N$**
>
>    • Inputs:
>
>    – ground-truth tokens $ \mathbf{x} _{1:t+n-1}$
>
>    – target-model features $ \mathbf{F} _{1:t}$ (unchanged)
>
>    – self-generated draft features $ \bar{\mathbf F} _{t+1:t+n-1}$ from the previous $(n−1)$ passes
>
>    • Outputs: token $ \bar{\mathbf x} _{t+n}$ and feature $ \bar{\mathbf F} _{t+n}$.
>
> Thus, with each additional drafted token, the proportion of **self-generated** draft features fed back into the draft model increases. This is what we mean by **gradually/progressively** shifting the model toward using its own outputs.
>
> **3. Step & Pass, and why acceptance lengths can be longer than step.**
> We thank the reviewer for raising this important question regarding the usage of the terms *step* and *pass*, and for pointing out the potential confusion related to Table 4 and Line 141.
>
> In Line 141, the variable *n* refers to the **index of the training step** in GRIFFIN’s progressive training schedule, rather than the total number of forward passes. Specifically, each training step *n* is designed to train the draft model to perform *n* consecutive forward passes, thereby predicting *n* future tokens and their corresponding features. This approach helps the draft model learn to operate under conditions that closely simulate those it will encounter during speculative decoding, where it must rely on its own previously generated tokens and features. For example, at training step 3, the draft model is trained to generate three tokens in a sequence, conditioning on the outputs of its own previous passes.
>
> It is also important to clarify why the **acceptance length** reported in Table 4 can exceed the value of the training step. The *step* index defines how the model is trained, not a hard constraint on how many tokens it can generate or how many the target model may accept during **decoding**. At inference, the draft model is allowed to generate a sequence of any length (e.g., >*N*), and the target model verifies them all at once. The **acceptance length** refers to the average number of draft tokens accepted by the target model per draft-verify cycle and is influenced by the quality of the generated tokens.  Accordingly, acceptance length can exceed the "step" value used during training.
>
> In our revised manuscript, we will explicitly clarify that "step" refers to the training schedule rather than a decoding limit, and we will add explanatory notes to Table 4 to prevent this misunderstanding. We appreciate the reviewer’s thoughtful feedback, which helps us improve the clarity and precision of our presentation.
>
> **4. Loss and Weight in Eqn 2.**
> We follow exactly the same dual-objective loss introduced in EAGLE. The complete definition of loss $\ell$ in Equation 2 is:
> $$
> \ell(\bar{\mathbf{x}}_t, \mathbf{x}_t, \bar{\mathbf{F}}_t, \mathbf{F}_t) = \ell_1(\bar{\mathbf{F}}_t, \mathbf{F}_t) + \omega * \ell_2(\bar{\mathbf{x}}_t, \mathbf{x}_t)
> $$
> where $\ell_1$ is the Smooth L1 distance and $\ell_2$ is the standard cross-entropy loss. For fair comparison, we follow EAGLE and HASS and set the weighted factor $\omega$ to 0.1.
>
> Because we did not tune $\omega$ and simply inherited the canonical setting from prior work, it was omitted from the hyper-parameter by oversight. We will provide full implementation details of the loss function in **Appendix C.2** of the revised manuscript.
>
> **5. Speedup Ratio and hardware setup.**  To clarify, the Speedup Ratio reflects the entire speculative decoding process, including both drafting and verification. All evaluation experiments were conducted on a single NVIDIA A100 80G GPU, except for LLaMA3-70B, which required two GPUs due to memory constraints. Both drafting and verification are performed on the same GPU, consistent with prior work (e.g., EAGLE, HASS), ensuring a fair comparison.
>
> Regarding the concern that speculative decoding might be slower than autoregressive decoding on a single GPU: this is a misconception. Although speculative decoding adds some overhead from drafting, it significantly reduces latency by verifying multiple tokens in parallel via a single forward pass of the target LLM model using batched causal masking. In contrast, autoregressive decoding requires one forward pass per token.
>
> This efficiency has been consistently validated in prior work (e.g., EAGLE, HASS), and our results confirm it: GRIFFIN achieves up to 18% speedup over EAGLE-2 and 7% over HASS, using the same GPU configuration. We will clarify this in the final version for clarity.

---

> > ### Comment · Reviewer_vyph · 2025-08-06
> >
> > Thanks for the detailed reply. Most of my concerns are addressed, except the last one—which I realize is a general question about speculative decoding. In any case, I’m raising my score to positive.
> >
> > >Speedup Ratio and hardware setup
> >
> > Could you share more details on how the KV cache is managed for the autoregressive baseline, drafting, and verification? My understanding is that with KV caching enabled, speculative decoding should have more FLOPs than the autoregressive baseline. Is this acceleration because of batch size = 1 and GPU is not compute or IO bound?  Also, does GRIFFIN still provide acceleration as batch size increases?

---

> ### Author Response · Authors · 2025-08-07
> **Clarification on Speedup Ratio and Hardware Setup**
>
> We greatly appreciate your thoughtful follow-up and your willingness to raise your score. Thank you for your careful reading and constructive engagement throughout the review process. Below, we provide a point-by-point response to address your concerns.
>
> **1. KV cache management.** Our implementation of KV cache for autoregressive decoding, drafting, and verification strictly follows the standard approaches used in prior works such as EAGLE-2 and HASS. Concretely, every decoding request maintains two independent caches:
>
> - *Target-model cache*: Persistently stores the key–value pairs of the target model throughout the conversation.
> - *Draft-model cache*: Holds keys and values generated by draft model; this cache is discarded immediately after the corresponding verification step and therefore does not accumulate over the entire sequence.
>
> Detailed implementation specifics can be found in our anonymous code repository (model/kv_cache.py).
>
> **2. FLOPs analysis.** Your intuition is correct: speculative decoding entails more *total* FLOPs than autoregressive decoding, since it invokes an extra draft model.  However, the draft model is typically less than 5% of the target model’s size, so the incremental FLOPs is minor.
>
> **3. Why speculative decoding achieves acceleration?** Your understanding of the reasons behind acceleration is correct. The primary reason speculative decoding can accelerate inference is that LLMs are not compute-bound during autoregressive decoding, especially when batch size are relatively small; during this stage, GPU utilization is relatively low. Speculative decoding improves GPU utilization by allowing the target model to verify multiple draft tokens in parallel within a single forward pass. Although the number of FLOPs per forward pass for the target model increases, the time required to verify and accept multiple tokens is comparable to the time needed to generate a single token in conventional autoregressive decoding.
>
> **4.GRIFFIN's performance at larger batch sizes.** To evaluate GRIFFIN’s performance under batch sizes greater than 1, we integrated GRIFFIN to **vLLM** following the same speculative decoding interface as EAGLE. We then measured throughput on **Llama-3-8B-Instruct** using the MT-Bench dataset (temperature $=0$) for batch sizes {$2,4,8,16$}.  Because the current vLLM interface does not yet support *tree-based* drafting, we evaluated all methods in a sequential setting with maximum chain length $=2$.  Results are summarised below (baseline $=1.00\times$ is vanilla vLLM decoding):
>
> | Batch Size | 2     | 4     | 8     | 16    |
> | :--------- | :---- | :---- | :---- | :---- |
> | EAGLE      | 1.37× | 1.32× | 1.28× | 1.18× |
> | HASS       | 1.40× | 1.35× | 1.30× | 1.20× |
> | GRIFFIN    | 1.52× | 1.45× | 1.37× | 1.25× |
>
> The observed trend—**declining speedup as batch size increases**—is theoretically expected. As batch size grows, the target model benefits from better GPU utilization, reducing computational redundancy and diminishing the marginal gain from speculative decoding. Furthermore, large batches begin to approach memory and compute bottlenecks, making the overhead from additional draft model calls more significant relative to the baseline. Nonetheless, GRIFFIN preserves a consistent **6–11%** advantage over EAGLE and **4–8%** over HASS across all tested batch sizes, demonstrating robust benefits even in high-throughput scenarios.

---

### Official Review · Reviewer_Cfb2 · 2025-06-23

**Clarity:** 2
**Significance:** 3
**Originality:** 2
**Rating:** 4
**Confidence:** 4

**Summary:**

This paper aims to address the feature and token misalignment issue in EAGLE. Specifically, during the training of EAGLE draft model, the draft model takes the feature and output tokens of the target model as inputs, but during inference it takes the feature and output tokens of the draft model as inputs. This misalignment results in less optimal accuracy for the draft model. To overcome this issue, it proposes (1) a new training method that discard training data that has mis-alignment issue; (2) propose a new module for the draft model to prioritize token embeddings over predicted features. Experiments over state-of-the-art baseliens confirm the effectiveness of the proposed method.

**Questions:**

1. Can authors help me with my confusion: I understand that the misalignment this paper aims to solve is that during the training of EAGLE draft model, the draft model takes the feature and output tokens of the target model as inputs, but during inference it takes the feature and output tokens of the draft model as inputs. But when designing the training mask (m_t), the value of m_t is determined by the prediction of step t, instead of the input of step t, which doesn't make sense to me. To my understanding, the m_t = 1 if the draft model makes a good prediction,  and it will be computed in the loss; and m_t = 0 if the draft model makes bad prediction (not in top-k prediction), and it will be discarded. So the training process only counted the training samples the draft model perform well? Why this approach can help address the misalignment issue? It seems more reasonable to use m_{t-1} to determine whether to include the loss in step t.

2. In the results of Table 1, there are many cases where the acceptance length is over 5 but the speed up ratio is only around 3. Can authors report the detailed breakdown of decoding latency of the draft and target model, and analyze what is the main bottleneck for the speed-up ratio?

3. I think running draft model for n-forward pass during training will slow down the training speed, perhaps quite significantly, which is also mentioned by the authors. Can authors report the training time comparision of the proposed method and the baseline?

4. I do not need authors to compare with EAGLE-3. But I want to discuss it with the authors. In EAGLE-3, they choose to drop the feature prediction in the draft model, which is better when the training data increases. I wonder how the authors think of their choice. Will the techniques proposed in this paper still be applicable to EAGLE-3?

**Ethical Concerns:**

["NO or VERY MINOR ethics concerns only"]

**Final Justification:**

The authors addressed my concerns. So I maintain the positive score.

**Limitations:**

The authors have discussed the limitation in their conclusion.

**Quality:**

3

**Strengths And Weaknesses:**

S1. I think the idea behind the proposed method is overall reasonable.

S2. The experiments cover state-of-the-art baselines and lots of base models. I believe the experiments are strong.

W1. The novelty is not excellent, but I believe it is still a solid work.

W2. I have some trouble understanding the design of token-alignable training (Sec 4.1). Please see my detailed questions below.

W3. I have some more questions about the details of the paper. Please see the questions below.

---

> ### Author Rebuttal · Authors · 2025-07-29
>
> Thank you for your thoughtful feedback. Below, we provide a point-by-point response to address your concerns.
>
> **1. Mask**. Thanks. The training mask $ \mathbf{m} _t$ was unfortunately misrepresented in our original text due to a typesetting oversight. The correct definition of Equation (3) should be $ \mathbf{m} _t = \prod _{i=t-n+1}^{t-1} \bar{\mathbf{m}} _i$. This formulation means that the loss mask at time step $t$ is determined by the **prediction quality of the previous $n-1$** draft tokens—not by the current prediction at $t$. Specifically, $ \bar{\mathbf{m}} _i = 1$ if the ground truth token at position $i$ is within the top-$k$ predictions of the draft model, and 0 otherwise. Therefore, $ \mathbf{m} _t = 1$ only if all previous draft tokens are well-aligned; otherwise, the loss at position $t$ is masked out.
>
> This design is motivated by the behavior of **speculative decoding**: once a misaligned token is encountered, **all subsequent draft tokens are rejected**. By mirroring this decoding logic during training, we ensure the model is only updated on token sequences that would have been **accepted in actual decoding scenarios**. In this way, the loss focuses on segments of the draft that contribute valid outputs, thereby improving the alignment between training and inference distributions.
>
> We will revise the equation and accompanying explanation in our manuscript for clarity.
>
> **2. Breakdown of Decoding Latency**. Thank you for the thoughtful question. The key reason the **speedup ratio** is lower than the **acceptance length** in Table 1 is due to the **non-negligible overhead of draft model inference**.
>
> To elaborate, let $N$ be the total number of tokens generated. Then the total latency for **autoregressive decoding** is $\mathbf{T}_a = N \cdot \mathbf{t}$, where $\mathbf{t}$ is the latency per forward pass of the target model. By comparison, the total latency for **speculative decoding** is $\mathbf{T}_s = \frac{N}{\tau} \cdot \left( \mathbf{t} + \mathbf{d} \cdot \bar{\mathbf{t}} \right)$,  where $\tau$ is the average number of tokens accepted per cycle, $\bar{\mathbf{t}}$ is the draft model's per-pass latency, and $\mathbf{d}$ is the draft tree depth. Thus, the **speedup ratio** becomes:
> $$
> \text{Speedup Ratio (SR)} = \frac{\mathbf{T}_a}{\mathbf{T}_s} = \frac{\mathbf{t}}{\mathbf{t} + \mathbf{d} \cdot \bar{\mathbf{t}}} \cdot \tau
> $$
>
> Using **LLaMA3-8B-Instruct on A100-80G** as a representative example, the latency per forward pass of target model is approximately **20-30ms**, for draft model is **1-2ms**.  So, if we set $\mathbf{t} = 25ms$, $\bar{\mathbf{t}} = 1.5ms$, $\tau = 5$ and $\mathbf{d} = 6$, we could roughly calculate the speedup ratio $SR = 3.68$ , which is closely matches our reported experimental results, confirming that the drafting overhead is the primary factor limiting speedup below the acceptance length. We will include a breakdown of draft vs. target model latency in the **Appendix H** in the revision.
>
> **3. Training time comparision**. GRIFFIN’s progressive training introduces some overhead due to multiple forward passes per training example. To quantify this, we compare the total training cost of GRIFFIN against HASS, a strong state-of-the-art baseline that also uses a multi-stage training schedule:
>
> | Model Size | HASS Training Cost      | GRIFFIN Training Cost   |
> | ---------- | ----------------------- | ----------------------- |
> | 7B         | ~140 A100-80G GPU hours | ~160 A100-80G GPU hours |
> | 13B        | ~240 A100-80G GPU hours | ~270 A100-80G GPU hours |
>
> While the training cost increases modestly, this overhead is amortized over large-scale deployment. Once trained, the draft model serves millions of inference requests. GRIFFIN yields up to 7% higher decoding speed over HASS at inference (Table 1), making it significantly more efficient in production.
>
> For example, in a system serving ~10,000 requests/day, let alone the billion requests/day of ChatGPTs and other popular LLMs, the additional training cost would be offset by inference gains within one week of deployment. We discuss this trade-off in detail in Appendix F, and believe the training overhead is well-justified by the lasting efficiency gains in real-world usage.
>
> **4. Discussion with EAGLE-3.** Thank you for opening this discussion about EAGLE-3’s decision to drop feature prediction loss in the draft model. To understand the practical impact of that choice and the possible interaction with our own techniques, we ran both (i) an **ablation on GRIFFIN that removes feature-level loss** and (ii) a head-to-head comparison with a faithfully re-implemented **EAGLE-3** draft model on LLaMA3-8B-Instruct.
>
> |         | MT-bench |          | HumanEval |          | GSM8K    |          | Average  |          |
> | ------- | -------- | -------- | --------- | -------- | -------- | -------- | -------- | -------- |
> | t=0     | $SR$     | $\tau$   | $SR$      | $\tau$   | $SR$     | $\tau$   | $SR$     | $\tau$   |
> | GRIFFIN | **3.09** | **4.85** | **3.65**  | **5.97** | **3.30** | **5.31** | **3.35** | **5.38** |
> | GRIFFIN + remove feature loss | 2.61     | 4.33     | 3.32      | 5.15     | 2.76     | 4.58     | 2.89     | 4.69     |
> | EAGLE-3 | 2.93     | 4.71     | 3.59      | 5.72     | 3.17     | 5.01     | 3.23     | 5.15     |
> | t=1     | $SR$     | $\tau$   | $SR$      | $\tau$   | $SR$     | $\tau$   | $SR$     | $\tau$   |
> | GRIFFIN | **2.62** | **4.35** | **3.31**  | **5.62** | **3.07** | **5.08** | **3.00** | **5.02** |
> | GRIFFIN + remove feature loss | 2.32     | 4.02     | 2.85      | 4.97     | 2.49     | 4.36     | 2.55     | 4.45     |
> | EAGLE-3 | 2.51     | 4.18     | 3.27      | 5.47     | 2.90     | 4.85     | 2.89     | 4.83     |
>
> - $SR$: Speedup Ratio; $\tau$ : Acceptance Length
>
> **EAGLE-3 Implementation Details**:
> For EAGLE-3, we used their official open-source codebase to train a draft model on the ShareGPT dataset. We made every effort to follow their protocol:
> - Training protocol and hyperparameters were kept fully consistent with those reported in the EAGLE-3 paper (e.g., learning rate schedule, optimizer, batch size, model configuration).
> - The only difference is the scale of training data: due to computational constraints, we used only ShareGPT (rather than the much larger UltraChat-200K + ShareGPT combination), as training on the full protocol would require approximately 2,400 A100-80G GPU hours—far beyond our resources for the rebuttal period. Both GRIFFIN and EAGLE-3 used identical ShareGPT training data, ensuring fair comparison.
>
> **Key observations**:
> - Removing feature-level loss from GRIFFIN causes a sizable drop (**10%-15%**) in both Speedup Ratio and Acceptance Length, confirming that feature-level loss is still important for our token-alignment objective.
> - Under identical ShareGPT training data and settings, GRIFFIN still **outperforms** EAGLE-3. This suggests that EAGLE-3’s “remove feature-level loss” design does NOT provide an advantage at this data scale.
>
> **Applicability of GRIFFIN techniques to EAGLE-3**:
> - EAGLE-3 does not explicitly address **token misalignment**, whereas GRIFFIN’s Token-Alignable Draft Model (TAD) and Token-Alignable Training (TAT) tackle this issue directly.
> - Both TAD and TAT can in principle be integrated with EAGLE-3-style draft models without altering their external interface.
>
> We will include a detailed discussion of these experimental results and their implications in **Appendix I** of our revised manuscript.

---

> > ### Comment · Reviewer_Cfb2 · 2025-08-01
> >
> > I appreciate the author's feedback. My concerns are addressed. I will keep my positive rating for this paper.

---

### Official Review · Reviewer_RdHq · 2025-06-27

**Clarity:** 3
**Significance:** 3
**Originality:** 4
**Rating:** 5
**Confidence:** 3

**Summary:**

The authors propose a modified framework for speculative decoding which compared to EAGLE and HASS (1) modifies the training by masking the loss for tokens where the prediction would not lie in the top-k, training the draft model to align with tree-based decoding, and by a modified draft model that has a Token-Guided Fusion (TGF) component on the input side merging information from the previous token features F_t and the current token x_t, and a Token-Enhanced Head (TEH) module which splits the draft model's hidden state into one F^p representation used for prediction and one f^R representation used as input in the next drafting token.
In experiments on MT-Bench, HumanEval and GSM8k using different Llama/Qwen/Mixtral models, the new approach consistently outperforms EAGLE-2, FSPAD and HASS on speedup ratio as well as acceptance length.

**Questions:**

While EAGLE-3 has come out after the ArXiv draft of this paper, it would be interesting to have a comparison if possible. Do you expect its performance to be roughly at the same level as HASS?

**Ethical Concerns:**

["NO or VERY MINOR ethics concerns only"]

**Final Justification:**

(8/3) After reading the comments and rebuttals I tentatively believe that the authors will be able improve the technical discussion in the paper to address the main points of the questions raised, but would be looking forward for the authors confirming this with a list of changes they want to make in the paper itself.

**Limitations:**

the case of batch sizes > 1 is not considered at all, which is in line with previous work
The authors mention higher training overhead compared to EAGLE, which I think is an interesting point that would otherwise be lost

**Quality:**

4

**Strengths And Weaknesses:**

Strengths:
- speculative decoding is a well-established task and the baselines provided are strong an meaningful
- empirical results show that the approach brings a useful and significant improvement
- the authors provide an ablation of the training loss and draft model parts of their contribution, showing how each novel element performs bz itself
- the authors promise to make their implementation and draft models available

Weaknesses:
 - the case of batch sizes > 1 is not considered at all, which is in line with previous work but also excludes a significant area of use-cases

---

> ### Author Rebuttal · Authors · 2025-07-29
>
> Thank you for your thoughtful feedback. Below, we provide a point-by-point response to address your concerns.
>
> **1. Batch sizes > 1**. To evaluate GRIFFIN’s performance under batch sizes greater than 1, we integrated GRIFFIN into **vLLM**, following the same speculative decoding interface as EAGLE. We then measured throughput on **LLaMA3-8B-Instruct** using the MT-bench dataset with temperature set to 0, across a range of batch sizes. As shown below, GRIFFIN consistently outperforms both EAGLE and HASS at all batch sizes:
>
> | Batch Size | 2     | 4     | 8     | 16    |
> | ---------- | ----- | ----- | ----- | ----- |
> | EAGLE      | 1.37× | 1.32× | 1.28× | 1.18× |
> | HASS       | 1.40× | 1.35× | 1.30× | 1.20× |
> | GRIFFIN    | 1.52× | 1.45× | 1.37× | 1.25× |
>
> Here, the baseline (1.00×) refers to standard vLLM decoding without any speculative decoding methods. It is important to note that these results were obtained under specific constraints imposed by vLLM’s current speculative decoding implementation. Specifically, the setup does **not support tree-based drafting**, so all evaluations used **sequential speculation** with a **maximum chain length of 2**. As a result, these results are not directly comparable to those reported in the main body of our paper, which uses a different decoding backend and configuration.
>
> The observed trend—**declining speedup as batch size increases**—is theoretically expected. As batch size grows, the target model benefits from better GPU utilization, reducing computational redundancy and diminishing the marginal gain from speculative decoding. Furthermore, large batches begin to approach memory and compute bottlenecks, making the overhead from additional draft model calls more significant relative to the baseline.
>
> Despite these challenges, GRIFFIN shows robust performance, delivering **6–11% higher throughput than EAGLE** and **4–8% higher than HASS** across all batch sizes. These results suggest that GRIFFIN’s advantages are preserved even under high-throughput, batch-inference settings, which are common in real-world deployments.
>
> We will include this analysis in **Appendix G** of the revised manuscript, along with: (1) full implementation details of the vLLM integration; (2) head-to-head comparisons with other speculative decoding methods in batch settings; and (3) a discussion of potential optimization strategies for improving speculative decoding under large-batch conditions.
>
> **2. Comparison with EAGLE-3.** We compared GRIFFIN with EAGLE-3 on **LLaMA3-8B-Instruct** across three benchmarks (MT-bench, HumanEval, GSM8K) at two temperature settings:
>
> |         | MT-bench |          | HumanEval |          | GSM8K    |          | Average  |          |
> | ------- | -------- | -------- | --------- | -------- | -------- | -------- | -------- | -------- |
> | t=0     | $ SR $     | $ \tau $   | $ SR $      | $ \tau $   | $ SR$     | $ \tau$   | $ SR$     | $ \tau$   |
> | GRIFFIN | **3.09** | **4.85** | **3.65**  | **5.97** | **3.30** | **5.31** | **3.35** | **5.38** |
> | EAGLE-3 | 2.93     | 4.71     | 3.59      | 5.72     | 3.17     | 5.01     | 3.23     | 5.15     |
> | t=1     | $ SR$     | $ \tau$   | $ SR$      | $ \tau$   | $ SR$     | $ \tau$   | $ SR$     | $ \tau$   |
> | GRIFFIN | **2.62** | **4.35** | **3.31**  | **5.62** | **3.07** | **5.08** | **3.00** | **5.02** |
> | EAGLE-3 | 2.51     | 4.18     | 3.27      | 5.47     | 2.90     | 4.85     | 2.89     | 4.83     |
>
> - $SR$: Speedup Ratio; $\tau$ : Acceptance Length
>
> Since EAGLE-3 doesn't provide pre-trained draft models for LLaMA3-8B, we used their official code to train on the ShareGPT dataset, maintaining all other hyperparameters consistent with their paper. Training EAGLE-3 on ShareGPT alone required over **300 A100-80G GPU hours**. Following their paper's full protocol (UltraChat-200K + ShareGPT) would require approximately **2,400 GPU hours**, which exceeded our rebuttal period resources. However, both methods used identical ShareGPT training data, ensuring fair comparison.
>
> GRIFFIN outperforms EAGLE-3 across all benchmarks and temperature settings. At $t=0$, GRIFFIN achieves **3.7%** higher speedup ratio and **4.5%** higher acceptance length compared with EAGLE-3. At $t=1$, GRIFFIN achieves **3.8%** higher speedup ratio and **3.9%** higher acceptance length compared with EAGLE-3.
> Interestingly, EAGLE-3's performance is **roughly comparable to HASS** levels, which aligns with your expectation. This suggests that while EAGLE-3 introduces architectural improvements, the performance gains are modest when training data is limited.
>
> We will include our experimental results of EAGLE-3 in the revised manuscript.

---

> > ### Comment · Reviewer_RdHq · 2025-08-01
> >
> > Thank you for the results and explanations on vLLM with batch size>1. Much appreciated!

---

### Official Review · Reviewer_wEUZ · 2025-07-02

**Clarity:** 3
**Significance:** 4
**Originality:** 4
**Rating:** 5
**Confidence:** 4

**Summary:**

This paper introduces a speculative decoding method, building on top of EAGLE, which improves the alignment of the draft model with the target model. This is done by: (i) training the draft model in stages of increasing number of forward passes, where in each stage the draft model is conditioned on its own generations rather than the teacher models generations; and (ii) changing the architecture of the draft model to increase the coupling between tokens and features.

Experimental results show significant improvements in speed and acceptance lengths over vanilla speculative decoding as well as EAGLE, across a variety of model sizes and eval benchmarks. Ablation studies establish the importance of both the training strategies, as well as the choice of hyperparameters.

**Questions:**

- Does the draft model need to be trained from scratch? Or can the architectural modifications be applied on top of existing pretrained models?
- How much training data is sufficient for updating the draft model?

**Ethical Concerns:**

["NO or VERY MINOR ethics concerns only"]

**Final Justification:**

As noted above, I think the paper is strong and the authors response addresses most of my comments. So I will recommend acceptance.

**Limitations:**

No comment

**Quality:**

3

**Strengths And Weaknesses:**

- Overall, the paper is technically strong -- it identifies an important limitation of prior SD methods, and proposes two detailed mechanims for combating that.
- The biggest strength of the paper lies in its empirical results, which show significant bumps in speed across many different SD setups. This makes the method practically useful.
- Ablation studies and thorough and clearly establish the importance of the proposed methods.

Weaknesses include:
- The architectural changes to the draft model may limit the applicability of the technique, and without the architectural change the stage-wise training strategy does not seem to be more effective than prior work.
- The paper is hard to follow for someone not super familiar with related works -- I had to go over the methods section multiple times. Adding conceptual summary before the technical details might help. Also the introduction mentions terms like "top-k", "forward n" without describing them clearly first.

---

> ### Author Rebuttal · Authors · 2025-07-29
>
> Thank you for your thoughtful feedback. Below, we provide a point-by-point response to address your concerns.
>
> **1. Architectural changes may limit the applicability**.  We would like to clarify that GRIFFIN’s draft model preserves identical input and output interfaces to those used in EAGLE and HASS, ensuring that our approach does not impose restrictive architectural modifications, at least in terms of input and output, and thus maintains broad applicability which means the same applicability as EAGLE and HASS.
>
> Moreover, as shown in Table 1, GRIFFIN consistently demonstrates strong performance across a diverse set of model families (LLaMA, Vicuna, Qwen, Mistral) and sizes (7B, 8B, 13B, 70B), confirming the robustness and generalizability of our method beyond any specific architecture or scale.
>
> Importantly, the two core components of our approach—Token-Alignable Training (TAT) and the Token-Alignable Draft Model (TAD)—are designed to be modular and independently integrable. TAT can be applied to any draft model training pipeline to effectively resolve input misalignment issues, while TAD’s architectural improvements can be adopted in existing draft model designs with minimal structural changes.
>
> **2. Adding conceptual summary for clarity.** Thank you for your valuable feedback.
> To improve clarity, we will add a **“Conceptual Overview”** section at the start of Section 3 in the revised manuscript. This overview will provide a high-level intuition behind the token misalignment problem, outline the logical flow from problem identification to our proposed solutions, and clearly map technical components to their respective roles.
>
> Additionally, we will carefully define key technical terms such as “top-k” and “forward n” upon their first introduction to ensure readers can follow the methods more easily.  We believe these changes will significantly enhance the paper’s readability and accessibility without sacrificing technical rigor.
>
> **3. Draft model trained from scratch?**  GRIFFIN’s draft model does indeed need to be **trained from scratch**, rather than fine-tuned from existing pretrained draft models. This is primarily due to the **architectural modifications** introduced in our Token-Alignable Draft Model (TAD), which alter the internal computation flow in a way that is not compatible with the weight initialization of standard pretrained draft models of EAGLE and HASS. However, training the draft model remains substantially less expensive than training full-scale language models, and we believe the efficiency gains during inference justify the added training cost.
>
> **4. Training data size for updating the draft model**.  In our experiments, we followed established practice by using the **entire ShareGPT dataset**, which aligns with the standard scale adopted in prior speculative decoding work such as EAGLE and HASS.
>
> Our current paper does not include a systematic analysis of the relationship between training data volume and draft model performance. This is primarily due to computational constraints—specifically, limited GPU availability during the rebuttal period—which prevented us from conducting large-scale data scaling experiments.
>
> With additional resources, we would pursue a more thorough investigation on this good direction, as we believe it is a critical step toward making speculative decoding stronger.

---

### Comment · Area_Chair_Lpje · 2025-08-01
**Request for clarifications**

Dear authors and reviewers, thank you for submitting your reviews and responses. After reading the paper and the reviews and responses, I would like to raise several additional questions that were mostly not covered in the reviews to help further clarify the technical aspects of the paper.

Method:

1. Thank you to reviewers Cfb2 and vyph for bringing up the mask indexing question, and thank you authors for explaining the mistake in equation 3 (see authors response to first question of reviewer Cfb2). I was also confused about this point when I read the paper. It seems like the same issue is present also in Eq. 2  and also in the writing (lines 160-161)? Can the authors please accurately clarify the mask indexing? Given that the masking method a main contribution of the paper, I am surprised that reviewers wEUZ and RdHq didn’t bring this up, and I ask them to re-validate their technical understanding of the paper.

2. Why is top-k used for masking? Speculative decoding acceptance has a probabilistic criterion that is based on the predicted token probabilities, so it is still possible for a drafted token outside of the top-k of the target model to be accepted.

3. Can you define the TEH module? I can guess how it works but I didn’t see any technical description and only a reference to [31] that doesn’t seem to have any module named TEH?

4. How is the design of TGF related to token alignment? Can you explain what components help specifically with alignment? Where there any experiments that demonstrated the benefit of this design specifically for addressing the token alignment aspect? Where any alternative baselines ablated with similar capacity (in terms of parameters/ flops)?

5. "we gradually shift the model toward using its own outputs during training" - Similar to reviewer vyph (q2) I am also confused by this phrasing. Specifically, can you clarify if there are any differences from the HASS method?

Experimentation setup:

1. How many total drafting steps are used in evaluation for all reported results?

2. Figure 1c - can you explain how the Misalignment Token Rate per forward pass was computed? Was it based on exact token match? Did forward pass 4 count only over cases where passes 1-3 of the draft were aligned?

Terminology and clarity:

1. The use of “outputs” in the paper and responses is sometimes ambiguous (e.g. “its own outputs”) since there are both token outputs and feature outputs.

2. Line 141 "each training step n involves draft model performing n forward passes to predict n future tokens” - similar to reviewer vyph (q3). I understand that “training step” here is likely following the HASS terminology which is unconventional in using it to describe the drafter’s lookahead steps (in ML this term is typically used for training progress). Does this suggest that the training steps are independent? For example, if you train the drafter to look 4 steps ahead the you need to do 1 + 2 + 3+ 4 = 10 forward passes during training? This seems redundant since you can reuse the state and simply do 4 forward passes for learning all 4 steps. Can you clarify?

3. Figure 3 seems to describe the decoding diagram rather than training. Given that it is referred  to from passages discussing training, it can be confusing a bit since it shows that the drafted token predictions are passed from step to step. I would suggest either including arrows with both decoding/ training like in Figure 2, or at least clarifying in the caption.

Thank you very much. I encourage the authors and all reviewers to participate in the discussion

---

> ### Author Response · Authors · 2025-08-05
> **Reply to "Request for clarifications" (1/4)**
>
> We would like to sincerely thank the Area Chair and the reviewers for your thorough and thoughtful feedback. We greatly appreciate the time and effort you have dedicated to evaluating our submission. Below, we provide a point-by-point response to address each of your concerns and clarify aspects of our work.
>
> ## 1. Mask Indexing
>
> **Clarification of the Mask–Indexing Scheme (Eq. 2, lines 160–161).**  We thank the AC and the reviewers for pointing out the imprecision in our exposition of the masking strategy. The essential point is that *token misalignment arises only from the second forward pass onward*.  Accordingly, the binary mask $\mathbf m_t$ must *not* influence the first–pass loss but becomes indispensable in later passes.
>
> **First Forward Pass ($n=1$).**  Because the draft model processes the identical prefix of ground truth tokens $ \mathbf x _{1:t-1}$ at both training and decoding time, all predicted tokens $ \bar{\mathbf{x}} _{t}$ are perfectly aligned.  The correct loss at step $t$ therefore reads:
>
> $$
> \mathcal{L} _{\mathcal{M}}^{(1)} = \ell(\bar{\mathbf{x}} _{t}, \mathbf{x} _{t}, \bar{\mathbf{F}} _{t}, \mathbf{F} _{t}),
> $$
>
> **$n$-th Forward Pass ($ n \ge 2 $).** From the second forward pass, during decoding, speculative decoding may reject a draft token $ \bar{ \mathbf x} _ {t}$, in which case *all* later tokens $ \bar{\mathbf x} _ {t+1}, \bar{\mathbf x} _ {t+2}, \ldots$ in that draft sequence are also discarded. So, during training, the subsequent draft tokens $ \bar{\mathbf x} _ {t+1},\bar{\mathbf x} _ {t+2},\ldots$ in this draft sequence are **misaligned tokens**. Training on those misaligned tokens provides no useful signal. Therefore, to prevent the model from being penalized for such inevitably rejected positions, we introduce a cumulative binary mask $ \mathbf m _ t = \prod _ {i=t-n+1}^{t-1} \bar{\mathbf m} _ i.$ The loss for the $n$-th forward pass is therefore:
>
> $$
> \mathcal{L} _ {\mathcal{M}}^{(n)} = \frac{1}{\sum _ {t=1}^{l} \mathbf{m} _ {t}} \sum _ {t=1}^{l} \mathbf{m} _ {t} \ell(\bar{\mathbf{x}} _ {t}, \mathbf{x} _ {t}, \bar{\mathbf{F}} _ {t}, \mathbf{F} _ {t}),
> $$
>
> **Manuscript revisions.**
>
> - Equation (2) will be corrected to the unmasked first-pass loss shown above.
> - A new equation will explicitly present the masked loss for $n\ge 2$.
> - Lines 160–161 will clarify that $ \mathbf m _t = 1$ for all positions in the first pass and becomes selective only in subsequent passes.
>
> These changes are notational only; the implementation and all reported results remain intact. We appreciate the reviewers’ careful reading and will incorporate the corrected equations and explanatory remarks in the final version.
>
>
>
> ## 2. Justification for the top-$k$ masking rule.
>
> We would like to clarify that the top-$k$ filter is applied to the draft model's logits—not the target model’s. Our key motivation is to ensure that the draft model is trained in a manner that is strictly aligned with its actual behavior during decoding.
>
> During speculative decoding, GRIFFIN follows the same draft tree expansion strategy as EAGLE-2. Specifically, at each speculation step, only the top-$k$ tokens from the draft model’s output are retained, while those ranked below the top-$k$ threshold are pruned. These lower-ranked tokens are never examined or evaluated by the target model and therefore have no influence on acceptance or rejection.
>
> To preserve this operational consistency, we adopt the same top-$k$ criterion during training by masking out tokens beyond the top-$k$ set in the loss computation. This ensures that the learning signal is focused exclusively on those draft tokens that could realistically be explored—and potentially accepted—by the target model during decoding. Thus, our masking strategy is deliberately chosen to align the draft model's training distribution with its decoding-time behavior, enhancing both efficiency and accuracy.

---

> ### Author Response · Authors · 2025-08-05
> **Reply to "Request for clarifications" (2/4)**
>
> ## 3. Definition of TEH Module
>
> Thanks. In the submission, we did not carefully introduced TEH module but only cited [31] due to two reasons: (1) TEH is not our technical contribution, as it is directly adapted from the design in [31]; and (2) given limited space, we prioritized detailing our novel components and omitted the full description of TEH. We will include the following more detailed explanation of TEH in the appendix of revision for completeness.
>
> To clarify, TEH corresponds to the decoupled projection head proposed in [31] (see Figure 5 in [31]). It is a compact and single-layer MLP that replaces the conventional output projection in the transformer decoder layer of the draft model.  Formally, in our context,  let $ \mathbf{g} \in \mathbb{R}^c$ denote the output of an autoregressive layer in our draft model (see Figure 3(a)), where $c$ is the intermediate size of autoregressive layer. TEH maps $ \mathbf{g}$ into two features via:
>
> $$
> [\bar{\mathbf{F}}^P, \bar{\mathbf{F}}^R] = \mathbf{g} \mathbf{W} _ {\text{TEH}} + \mathbf{b} _ {\text{TEH}}
> $$
>
> where $ \mathbf{W} _ {\text{TEH}} \in \mathbb{R}^{c \times 2d}$ and $\mathbf{b} _ {\text{TEH}} \in \mathbb{R}^{2d}$ are the projection weights and bias, and $d$ is the model’s hidden size. Here $ \bar{\mathbf{F}}^P \in \mathbb{R}^d$ is fed to the LM head for token prediction; while $ \bar{\mathbf{F}}^R \in \mathbb{R}^d$ serves as the recurrent feature, passed to the next layer for feature propagation.
>
> The core motivation of TEH is to explicitly decouple the two objectives of decoding: accurate token prediction and stable hidden state propagation. A shared representation often leads to a conflict between these objectives, especially in draft models with minimal recurrence. By introducing separate projections, TEH improves the robustness of recurrence without sacrificing prediction quality [31].
>
>
>
> ## 4. TGF
>
> **Relation between TGF and token alignment.** TGF module is explicitly designed to enhance the draft model’s token prediction accuracy, which in turn reduces the number of misaligned tokens during token-alignable training.
>
> TGF module is motivated by the limitations of the standard concat-then-MLP strategy, as adopted in EAGLE, which does not fully capture the complementary information between token embeddings and draft model features. In practice, features generated by the draft model often remain misaligned with the target model’s representations, a discrepancy that cannot be effectively eliminated with feature-level loss minimization alone. TGF addresses this challenge by explicitly leveraging token embeddings to guide the fusion process, aligning feature distributions more closely to those of the target model. As confirmed by ablation results (Table 5), this targeted architectural enhancement significantly reduces feature inconsistency, demonstrating that the modest complexity introduced by TGF provides strong empirical gains.
>
> Empirically, the reduction in misalignment rate increases the proportion of draft tokens that pass the alignment mask, effectively expanding the usable training signal and improving overall learning efficiency. Therefore, TGF plays a direct role in boosting alignment success through improved token quality.
>
> **Which components foster token alignment in TGF?** In Table 6  of Appendix A.2, we have conducted a comprehensive ablation study, where we selectively removed each component of TGF while maintaining comparable parameter counts  and FLOP budgets. The key observations are:
>
> - **Removing both the Up-Projector and SiLU** lowers acceptance length by $0.23$ ($T{=}0$) and $0.27$ ($T{=}1$), and reduces speed-up by $0.17$ and $0.21$, respectively.
>
> - **Omitting the token embedding $\mathbf x$** from the second fusion step degrades acceptance length by $0.10$ ($T{=}0$) and $0.13$ ($T{=}1$), with speed-up dropping by $0.07$ and $0.08$.
> - **Excluding the recurrent feature $\mathbf h$** yields the most pronounced decline: acceptance length falls by $2.63$ ($T{=}0$) and $2.49$ ($T{=}1$), while speed-up plummets by $1.50$ and $1.39$.
>
> These controlled ablations demonstrate that each component of TGF plays a non-trivial role in improving the quality of draft tokens that ultimately pass verification. In particular, the recurrent integration of the fused feature $\mathbf{h}$ proves essential for capturing high-fidelity, alignable representations. Importantly, we did not identify any alternative architecture with similar parameter/FLOP budgets that matched TGF’s performance, underscoring the necessity and effectiveness of its specific design choices in addressing the token alignment challenge.

---

> ### Author Response · Authors · 2025-08-05
> **Reply to "Request for clarifications" (3/4)**
>
> ## 5. Difference with HASS
>
> **Clarifying “gradually shifting the model toward using its own outputs’’.** The quoted phrase refers to a *progressive self-conditioning mechanism* where the draft model increasingly relies on its own generated features rather than target features during the training process.
>
> For example, at *training step* $ N \ge 2$, the draft model performs exactly $N$ forward passes to predict the sequence of draft tokens $ \bar{\mathbf x} _ {t+1:t+N}$.
>
> 1. **Forward pass 1 (predict $ \bar{\mathbf x} _ {t+1}$):**
>
> - Inputs: ground-truth tokens $ \mathbf{x} _ {1:t}$ and target-model features $ \mathbf{F} _ {1:t}$.
> - Outputs: token $ \bar{\mathbf x} _ {t+1}$ and feature $ \bar{\mathbf F} _ {t+1}$.
>
> 2. **Forward pass $n$ (predict $ \bar{\mathbf x} _ {t+n}$), $2 \leq n \leq N$:**
>
> - Inputs:
>   - ground-truth tokens $ \mathbf{x} _ {1:t}$ (unchanged).
>   - target-model features $ \mathbf{F} _ {1:t}$ (unchanged).
>   - self-generated draft features $ \bar{\mathbf F} _ {t+1:t+n-1}$ from the previous $ (n-1)$ passes.
>   - self-generated draft tokens $ \bar{\mathbf x} _ {t+1:t+n-1}$ from the previous $ (n-1)$ passes.
> - Outputs: token $ \bar{\mathbf x} _ {t+n}$ and feature $ \bar{\mathbf F} _ {t+n}$.
>
> We can observe that along with $n$ increases, the draft model uses less ground-truth tokens $ \mathbf{x} _ {1:t}$ and target-model features $ \mathbf{F} _ {1:t}$ as the input, but takes more self-generated draft tokens  $ \bar{\mathbf x} _ {t+1:t+n-1}$ and draft features $ \bar{\mathbf F} _ {t+1:t+n-1}$ from the previous $(n-1)$ passes. This is what we mean by *gradually shifting the model toward using its own outputs*.
>
> **How HASS differs.** While GRIFFIN and HASS share similar training schemas, the **misalignment** each addresses is fundamentally different. As illustrated in Fig. 2,
>
> - **Feature misalignment** (addressed by HASS) occurs because during decoding, the draft model uses draft features $ \bar{\mathbf{F}} _ t$ to generate $ \bar{\mathbf{F}} _ {t+1}$, while during training it uses target features $ \mathbf{F} _ t$.
> - **Token misalignment** (GRIFFIN focus) arises because during decoding, the draft model uses draft tokens $ \bar{\mathbf{x}} _ t$ to generate $ \bar{\mathbf{F}} _ {t+1}$, while during training it uses ground truth tokens $ \mathbf{x} _ t$.
>
> HASS resolves *feature misalignment* by aligning the inputs at the feature level—specifically, it replaces $ \mathbf F _ t$ with $ \bar{\mathbf F} _ t$ during training.  *Token misalignment*, however, cannot be fixed by a naive substitution $ \mathbf x _ t \rightarrow \bar{\mathbf x} _ t$ because we operate under the fixed-dataset regime established by EAGLE and HASS (Sec. 3, lines 124–131).  The core issue is as follows:
>
> Frameworks such as EAGLE and HASS precompute and store $ \mathbf{F} _ t$ for all $ \mathbf{x} _ t$ *before* training begins, to avoid the computational cost of regenerating training data on the fly. As a result, naively replacing $ \mathbf{x} _ t$ with $ \bar{\mathbf{x}} _ t$ produces mismatched input-feature pairs:
>
> - The precomputed target feature $ \mathbf{F} _ {t+1}$ was computed on the ground-truth tokens $ \mathbf{x} _ {1:t}$ using the target model;
> - The corresponding draft feature $ \bar{\mathbf{F}} _ {t+1}$, however, would be computed using tokens $ \mathbf{x} _ {1:t-1}$ concatenated with the self-generated token $ \bar{\mathbf{x}} _ t$.
>
> Consequently, the feature-level loss $ \ell(\mathbf{F} _ {t+1}, \bar{\mathbf{F}} _ {t+1})$ becomes ill-defined since the two features are conditioned on different token inputs. As confirmed in Appendix B of HASS, this naive substitution undermines training and ultimately degrades final model performance.
>
> **Our solution: token-alignable training and draft model.** GRIFFIN not only preserves the strengths of HASS in addressing feature misalignment, but also introduces two complementary mechanisms specifically tailored to token misalignment:
>
> - **Token-alignable training.** A loss-masking scheme discards highly misaligned positions, preventing their erroneous gradients from corrupting learning.
> - **Token-alignable draft model.** Architectural modifications explicitly incorporate input tokens twice (initially with features and later to refine them) to reconcile discrepancies in the generated draft features.
>
> By jointly employing token-alignable training and a token-alignable draft model, GRIFFIN robustly addresses token misalignment while fully retaining the feature-level alignment advances established by HASS. This comprehensive approach yields both improved training stability and superior model performance, as substantiated in our empirical results in Table 1.

---

> ### Author Response · Authors · 2025-08-05
> **Reply to "Request for clarifications" (4/4)**
>
> ## 6. Draft step in Evaluation
>
> For comparability with HASS and EAGLE-2, we follow their settings and adopt identical hyper-parameters throughout our experiments.  Specifically, the total number of draft tokens per draft–verify cycle is fixed to $60$ for 7B/8B models, $50$ for 13B models, and $48$ for 70B models. These tokens are generated across a draft-tree of depth $6$, i.e., the draft model executes six successive draft steps before the verifier is invoked.
>
>
>
> ## 7. Calculation of Misalignment Token Rate
>
> Misalignment Token Rate (MTR) is calculated based on the top-$k$ alignment mask $ \mathbf{m} _ t$:
>
> $$
> \text{MTR} = \frac{\sum _ {t=1}^{l}\mathbf{m} _ {t}}{l}
> $$
>
> where $l$ denotes the sequence length of training data and $ \mathbf{m} _ t$ denotes the alignment mask.
>
> In Fig. 1(c) we report $\text{MTR}$ averaged over the entire ShareGPT training corpus for each forward pass. Crucially, the rate for pass 4 is computed over **all** tokens in the dataset; it is **not** exclusively over cases where passes 1-3 of the draft tokens were aligned.
>
>
>
> ## 8. Clarification of the term “outputs.”
>
> We acknowledge that the manuscript occasionally uses the word *outputs* without specifying whether it refers to *token outputs*  or *feature outputs*. In the revised version, **Token outputs** and **feature outputs** will be used only when the context already makes the level (token vs. feature) explicit; otherwise we will employ the full expressions above.
>
>
>
> ## 9. Clarification of Training Steps
>
> Thanks. To clarify, our use of “training step” follows the terminology introduced in HASS and refers specifically to the lookahead depth (the number of forward passes for draft model).
>
> Each training step is independent in terms of computation. Specifically:
>
> - At training step $n$, the draft model is trained to perform exactly $n$ forward passes to predict $n$ future tokens.
> - We do not accumulate passes from previous steps. That is, training for Step 4 requires only 4 forward passes—not 1+2+3+4 =10.
>
> Training proceeds in stages as follows:
>
> - **Step 1. Initialize from $\mathcal{M}_0$ and train for $1$-step to obtain $\mathcal{M}_1$:** Starting with initialization parameters $\mathcal{M}_0$, we train the draft model on the entire dataset to perform a *single* forward pass per position, yielding a new model checkpoint $\mathcal{M}_1$.
> - **Step 2. Train $\mathcal{M}_1$ for $2$-step to obtain $\mathcal{M}_2$:** Using $\mathcal{M}_1$ as initialization, we now train the draft model (over the entire dataset) to perform *two* forward passes per position, resulting in $\mathcal{M}_2$.
> - **Step $n$. Train $\mathcal{M}_{n-1}$ for $n$-step to obtain $\mathcal{M}_n$:** Recursively, at training step $n$, we initialize from $\mathcal{M}_{n-1}$ and train the draft model to conduct *exactly $n$ forward passes* per position, producing the updated model $\mathcal{M}_n$.
>
> Thus, during the curriculum phase corresponding to training step $n$, the model is run **exactly $n$** forward passes per sequence, *not* the cumulative sum $1{+}\dotsb{+}n$.  This design provides a stable, progressively harder training curriculum: each subsequent step starts from a well-converged checkpoint and adapts it to a deeper draft length.
>
>
>
> ## 10. Clarfication of Fig. 3.
>
> Figure 3 depicts the *architecture* of the proposed *token-alignable draft model*, not a standalone training diagram.  Because this architecture is shared between training and decoding, the figure necessarily shows drafted tokens flowing from one step to the next; such arrows are valid in both regimes. To pre-empt confusion, we will revise the caption to state explicitly that the illustration applies to *both* training and decoding.

---

> > ### Comment · Area_Chair_Lpje · 2025-08-06
> >
> > Dear authors,
> >
> > Thank you very much for your detailed response to help clarify the presented method. Especially answer (9) explains the set up and helps contextualize the rest of the details. Given the many different components, I find it important to be accurate in all the descriptions. One more small thing: in equation (4) in the paper, should the subscript indices be up to $t-1$ instead of $t$?
> >
> > Another followup question on the details of the implementation: For using the self-generated tokens, are you using greedy sampling?

---

> > > ### Author Response · Authors · 2025-08-06
> > > **Reply to the Area Chair’s follow-up questions**
> > >
> > > We sincerely thank the Area Chair for the careful feedback. Below, we address each of your concerns in detail and clarify specific aspects of our work point by point.
> > >
> > > **1. Corrected indices in Equation. (4).** Yes, the subscripts in Equation. (4) should run only up to $t-1$.
> > > The corrected equation reads
> > >
> > > $$
> > > \begin{equation}
> > > 	\begin{aligned}
> > > 		\bar{\mathbf{F}} _ {t}= \mathcal{M}(\mathbf{x} _ {1:t-1}, \mathbf{F} _ {1:t-n}, \bar{\mathbf{F}} _ {t-n+1:t-1}),  \quad
> > > 		 \bar{\mathbf{x}} _ {t}  = \mathcal{H}(\bar{\mathbf{F}} _ {t}). \\
> > > 	\end{aligned}
> > > \end{equation}
> > > $$
> > >
> > > We will amend this typo in the revised manuscript.
> > >
> > > **2. Sampling strategy for self-generated tokens.**
> > > Following other works like EAGLE-2 and HASS for fair comparison, during both training and decoding, we adopt *top-$k$ sampling*, not greedy sampling. Concretely,
> > >
> > > - *Decoding:* GRIFFIN follows the same draft tree expansion strategy as EAGLE-2. The draft model samples from its top-$k$ draft tokens at each position to grow a draft tree, which is subsequently verified by the target model.
> > > - *Training:* We apply the same top-$k$ constraint by masking draft tokens outside the top-$k$ set in the loss computation. This guarantees that the learning signal is restricted to draft token predictions that could plausibly arise—and be accepted—at decoding time.

---

### Note · Authors · 2025-08-13

> For clarity, we refer to the reviewers as follows: R1 (wEUZ), R2 (RdHq), R3 (Cfb2), and R4 (vyph).

We sincerely thank the Reviewers and the Area Chair for their careful reading and constructive feedback. All reviewers recognized the novelty and importance of our work, highlighting GRIFFIN's token-alignable training and draft model are well motivated and technically strong (R1, R2, R3, R4), the empirical improvements of GRIFFIN are significant (R1, R2, R3, R4), and the ablation studies clearly establish the importance of the proposed methods (R1, R2).

The primary concerns raised during the initial reviews were systematically addressed in our rebuttal through targeted clarifications and additional experiments. Specifically: (1) performance at batch-size > 1 (R2), we provided vLLM-based results demonstrating GRIFFIN's consistent advantages, which R2 acknowledged with appreciation; (2) masking implementation details and comparison with EAGLE-3 (R3), we offered detailed technical explanations and ablation experiments with EAGLE-3 that fully resolved R3's concerns; and (3) the goal of token alignment (R4), our clarifications successfully addressed R4's questions. Additionally, we expanded our discussion of EAGLE-3 (R3's suggestion) to provide stronger justification for our draft-model design choices.

Beyond the reviewer-specific concerns, the Area Chair requested detailed clarifications of several technical aspects, including the definition of the TEH module, the scheduling mechanism of "Training Step" and so on. We provided comprehensive, point-by-point responses to each inquiry. The Area Chair acknowledged that these explanations effectively clarified the experimental setup and provided essential context for understanding the technical details.

Following our comprehensive rebuttal, R4 raised the score to positive, and R1, R2, and R3 maintained their consistent acceptance recommendations throughout the review cycle. The reviewers collectively endorsed our core contributions—the token-alignable training and draft model—as both well-motivated and empirically validated.

We are deeply grateful to all reviewers and the Area Chair for their meticulous attention to detail and constructive suggestions, which have substantially improved the quality and clarity of our work. We are committed to incorporating all these clarifications and improvements into the revised manuscript to ensure the highest standards of scientific presentation.

---

### Decision · Program_Chairs · 2025-09-17

**Decision:**

Accept (poster)

**Comment:**

This paper proposes GRIFFIN, which improves on previous self-drafting techniques such as EAGLE which are used for inference time speedup with speculative decoding via drafting and verification. The authors study the "token misalignment" problem, where the draft model is trained on ground-truth tokens that are being paired with actviations from previuos drafting steps that might have predicted different tokens. The authors proposed to address this alignment by modifying the training to do actual samling from the drafter. Noe that this makes the training more expensive since it allow teacher forcing and the authors use a curriculum where each epoch the drafter samples gradually more steps, but only the prediction from the last step in each epoch is used for the loss computation. In addition, the authors also propose to mask out trajectories after a misalignemnt event between the drafter and main model (defined by top-k), claiming that future tokens in this trajectory might introduce unnecesary signal. Finally, the authors add a “token-alignable draft” (TGF) module that through ablations they find to help improve the drafting accuracy without increasing the drafting inference-time latency, leading to overall inference speedups compared to existing baselines.

Strengths:
1. The empirical results are a major strength of the paper, improving over previous EAGLE/ HASS methods (comparison to EAGLE-3 was added during the rebuttal)
2. Studying and discussing the token alignemnt challenge in self speluclation models. While teacher-forcing works well for LLMS for enabling efficient training, in the studied architectures there is the added element of concatenating previous activations with token embeddings, which is worth studying further.
3. Positive scaling results

Weaknesses:

The main weakness of the paper is its current presentation that is unclear (even for readers with relevant background) and imprecise. The paper must be revised to carefully address all the points discussed below, including:
1. Going over all equations and notations to make sure they accurately describe the implementation.
2. Provide enough technical details to accurately describe all the techniques discussed in the paper
3. Clearly position the paper against other related works such as HASS and EAGLE, and in the method section describe which parts are adopted and which parts are new to this paper.
4. Be careful about terminology and notation, especically when using terms like “outputs” or “predictions” in this context- to be clear about what is meant.
5. Fully state all details of the the evalution setup

Other weaknesses of the proposed method are mostly around the more expensive training due to avoiding teacher forcing in favor of sampling, some gaps in claims like whether TGF truly helps specifically via token alignemnt as claimed or just generally helps generally with accuracy in other ways, ablations on whether masking helped or not, and justification for using top-k for masking.